# Glauber Generative Model: Discrete Diffusion Models via Binary Classification

**Harshit Varma** [*]
Inception Labs
`harshit@inceptionai.xyz`

**Dheeraj Nagaraj**
Google DeepMind
`dheerajnagaraj@google.com`

**Karthikeyan Shanmugam**
Google DeepMind
`karthikeyanvs@google.com`

## Abstract

We introduce the Glauber Generative Model (GGM), a new class of discrete diffusion models, to obtain new samples from a distribution given samples from a discrete space. GGM deploys a discrete Markov chain called the heat bath dynamics (or the Glauber dynamics) to denoise a sequence of noisy tokens to a sample from a joint distribution of discrete tokens. Our novel conceptual framework provides an exact reduction of the task of learning the denoising Markov chain to solving a class of binary classification tasks. More specifically, the model learns to classify a given token in a noisy sequence as signal or noise. In contrast, prior works on discrete diffusion models either solve regression problems to learn importance ratios, or minimize loss functions given by variational approximations. We apply GGM to language modeling and image generation, where images are discretized using image tokenizers like VQGANs. We show that it outperforms existing discrete diffusion models in language generation, and demonstrates strong performance for image generation without using dataset-specific image tokenizers. We also show that our model is capable of performing well in zero-shot control settings like text and image infilling.

## 1 Introduction

Diffusion Models (Sohl-Dickstein et al., 2015; Ho et al., 2020; Song et al., 2020) use continuous time, continuous space diffusion processes to sample from a target distribution. These models start with pure noise and learn to 'denoise' until a meaningful sample is produced and are very successful in generative modeling of images (Rombach et al., 2021; Patil et al., 2022; Saharia et al., 2022; Ramesh et al., 2021). These models are trained by solving a class of regression tasks called score matching (Hyvärinen & Dayan, 2005). Another popular choice is the class of auto-regressive models based on the transformer architecture, which are state-of-the-art in generative modeling of languages (Vaswani et al., 2017; Kenton & Toutanova, 2019; Brown et al., 2020; Team et al., 2023; Esser et al., 2021; Yu et al., 2022). These models work with a discrete set (the set of 'tokens'), where sequences of tokens make up meaningful text. They learn to generate tokens autoregressively by solving a multi-class classification problem.

Our work considers discrete diffusion models, where the model learns to denoise a sequence of tokens to produce a sample from a given distribution over token sequences. In language generation, this can mean starting from gibberish and denoising it to obtain a meaningful sentence. Such models have been studied in the literature to try and combine the successes of Diffusion models and Autoregressive models. We propose a new class of discrete diffusion models called the Glauber Generative Model (GGM) where the the denoiser is a discrete time Markov chain over token sequences, called the Glauber dynamics (also known as heat bath dynamics and Gibbs sampling). This is a central object of study in Statistical Physics of spin systems, Probability Theory and Bayesian inference

---

[*] work done while employed at Google DeepMind.

(Martinelli, 1999; Martinelli & Olivieri, 1994; Levin & Peres, 2017; Gelfand & Smith, 1990; Geman & Geman, 1984; He et al., 2016) and is a discrete analogue of Langevin Dynamics. GGM is trained by solving a well defined class of binary-classification problems – one of the simplest machine learning tasks – to classify if a specific token in a sequence of noisy tokens is 'signal' or 'noise'. In the context of language modeling, this roughly means figuring out whether a word is out-of-place in a sentence. We show an exact reduction from such a classifier to a denoising Glauber dynamics based sampler in Section 3.

## 1.1 RELATED WORK

Recent works have explored diffusion models for discrete data. Approaches for diffusion on discrete data can be broadly classified into two kinds: (i) discrete diffusion in the token space where a discrete time Markov chain is the denoiser, (ii) continuous diffusion in an embedding space of the discrete data. We briefly discuss these approaches and their applications in this section.

**Discrete diffusion via Discrete Markov Chains:** Diffusion models over discrete spaces, analogous to continuous space diffusion, were introduced in (Sohl-Dickstein et al., 2015) for generative modeling. Argmax flows (Hoogeboom et al., 2021) and D3PM (Austin et al., 2021) (and refinements such as (Zheng et al., 2023)) solidified these ideas – by considering the noising process to be a discrete time, discrete space Markov chain. They propose a variational loss function to teach a neural network to reverse this Markov chain. More specifically, D3PM considers character-level language modeling, token-level language modeling on short sequences (128) with relatively small vocabularies (8192), and low-resolution (e.g., $32 \times 32$) image generation tasks with small models (e.g., 36M)). On language modeling tasks these models still perform worse than an autoregressive transformer-based model with comparable parameters. Scaling these models to larger vocabularies is an active area of research. MDLM (Sahoo et al., 2024) simplifies and improves upon D3PM by making different engineering choices and considering a simplified training objective for a specific noising process. Using image tokenizers like VQGANs (Esser et al., 2021), VQ-DDM (Hu et al., 2021) and VQ-Diffusion (Gu et al., 2021) are able to apply models similar to Argmax flows and D3PM to generate images with larger resolutions. Within this framework, DiffusER (Reid et al., 2022) considers the noising Markov Chain to be Levenshtein edit operations over text for language modeling and evaluates its performance for downstream tasks such as translation and textual style transfer.

While continuous diffusion models are learnt by solving the regression task of score matching, another line of work (Lou et al., 2023; Meng et al., 2022) considers its discrete analogue called ratio matching (Hyvärinen, 2007; Sun et al., 2022). These works that argue that Argmax flow and D3PM in fact learn these importance ratios of probability distributions indirectly, and propose to learn these directly by solving a class of regression tasks. While (Meng et al., 2022) considers least squares regression in this setting, (Lou et al., 2023) uses a tailored loss function. Directly learning these ratios improves the learning performance as observed in SEDD (Lou et al., 2023), achieving competitive performance to autoregressive models for language modeling via discrete diffusion.

**Discrete Diffusion via Continuous Diffusion over Embeddings:** This line of work, introduced by (Li et al., 2022) and further explored in (Gulrajani & Hashimoto, 2023; Strudel et al., 2022; Ye et al., 2023; He et al., 2022; Yuan et al., 2022) considers embedding the discrete space in a continuous space and applies continuous diffusion models. Most notably, Plaid (Gulrajani & Hashimoto, 2023) (1.3B parameters) outperforms a 124M GPT2 model in likelihood on several language modeling benchmarks. However, GPT2-medium with 345M parameters ($3.8\times$ less parameters than Plaid) still outperforms Plaid on all of the benchmarks used in the evaluation – indicating a need for further improvement. Approaches like SSD-LM (Han et al., 2022) and TESS (Mahabadi et al., 2023) apply continuous diffusion in a logit space constructed over the vocabulary and evaluate on downstream tasks such as question generation and summarization.

**Applications:** Diffusion-based approaches for modeling discrete data have also been utilized in several recent applications. DiMA (Meshchaninov et al., 2024) uses continuous diffusion in the embedding space of a protein language model and outperforms autoregressive language models at unconditional protein sequence generation. DFMs (Campbell et al., 2024) propose a class of flow-based models designed for discrete data that achieve state-of-the-art results for protein co-design.

**Glauber Dynamics:** (also known as Gibbs Sampling) has been studied extensively in the statistical physics literature to understand spin systems (Martinelli & Olivieri, 1994; Martinelli, 1999). The study of its mixing properties for various spin systems is an important sub-field of probability theory (Levin & Peres, 2017; Benjamini et al., 2005; Diaconis & Ram, 2000; Mossel & Sly, 2010; Gheissari & Sinclair, 2022; Guo & Jerrum, 2017). Gibbs sampling has found numerous applications in Statistics and Bayesian Machine Learning for sampling from discrete posterior distributions (Gelfand & Smith, 1990; Geman & Geman, 1984; Zhang et al., 2001; Zhu et al., 1998).

## 1.2 OUR CONTRIBUTIONS

We introduce a novel theoretical framework for training discrete diffusion models, called GGM, based on an exact reduction to solving a class of $O(T|\mathcal{X}|)$ binary classification problems where $\mathcal{X}$ is the set of tokens and $T$ is the number denoising steps. Prior works based on D3PM incur a complexity of $O(|\mathcal{X}|^2 T)$ in general to learn the denoising Markov chain directly (see Remark 1). Moreover, the denoising process of GGM is a time dependent Markov chain which flips one token at a time, in contrast with prior works which flip multiple tokens simultaneously. Our empirical evaluation shows that GGM obtains a strong performance in the case of language generation, where it outperforms existing discrete diffusion models (Table 1). We show that time-independent Glauber dynamics implemented with masked language models like BERT cannot attain such a performance even with $32\times$ more steps (Figure 2a). We also demonstrate GGM's ability to generate high quality images on $256 \times 256$ CelebA-HQ and FFHQ datasets, where its performance rivals several popular diffusion models and GANs, without using a dataset-specific image tokenizer. We then demonstrate that our model can perform zero-shot image and text infilling with a variety of different masks. For the task of image generation, all the state-of-the-art methods use dataset-specific tokenizers and additional optimizations. We believe such techniques can further boost our model's performance and bridge the gap between our models and the state-of-the-art. Thus, we believe our framework is conceptually elegant, scalable, empirically competitive, and has the potential to be widely used for generative modeling.

## 2 PROBLEM SETUP

### 2.1 NOTATION

By $\mathcal{X}$ we denote a discrete set and call its elements as 'tokens'. Given $L \in \mathbb{N}$, we let $\mathcal{X}^L$ denote $\mathcal{X}$ valued sequences of length $L$. Given any $x \in \mathcal{X}^L$ and $i \in \{0, \ldots, L-1\}$, $x_{-i} \in \mathcal{X}^{L-1}$ denotes the sequence of length $L-1$ obtained form $x$ when the $i$-th position is removed. $x_i \in \mathcal{X}$ denotes the $i$-th position of $x$. Given a finite set $A$, we let $\mathsf{Unif}(A)$ denote the uniform probability distribution over the elements of $A$. For a probability distribution $Q$ over a finite space $\mathcal{Y}$, with $A \subseteq \mathcal{Y}$, $Q(|A)$ denotes its conditional distribution over $A$.

### 2.2 GLAUBER DYNAMICS

Let $\mathcal{X}$ be a finite set (such as all possible tokens in a language model or in a VQGAN). Glauber dynamics is a Markov chain to sample a joint distribution over the space $\mathcal{X}^L$ by flipping one token at a time. Given a probability distribution $P$ over $\mathcal{X}^L$, and $X_0 \in \mathcal{X}^L$, Glauber dynamics obtains the trajectory $X_0, \ldots, X_T$ by sampling $X_{t+1}$ given $X_t$ as follows:

1. Sample $I_t \sim \mathsf{Unif}(\{0, 1, \ldots, n-1\})$ independent of $X_t$.
2. $X_{t+1,i} = X_{t,i}$ if $i \neq I_t$.
3. $X_{t+1,i} \sim P(X_i | X_{-i} = X_{t,-i})$ if $i = I_t$

Glauber dynamics has $P$ as its unique stationary distribution under mild conditions over $P$. Even with exact knowledge of $P(X_i | X_{-i} = X_{t,-i})$, the iterates of Glauber dynamics can be very slow to converge to the distribution $P$, often requiring time exponential in $L$. In this work we consider Glauber dynamics with two differences:

1. $I_t$ is chosen from a fixed permutation in a round robin fashion (as in (He et al., 2016)).
2. The dynamics is time dependent. That is, $P(X_i | X_{-i} = X_{t,-i})$ is replaced by $P_t(X_i | X_{-i} = X_{t,-i})$ for $t = 0, 1, \ldots, T-1$ for some carefully designed $P_t$.

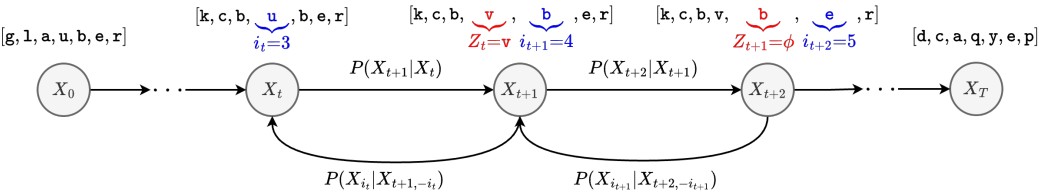

Figure 1: Example of Glauber dynamics in a discrete token space, where the tokens are characters.

In this framework, we obtain an exact sample from $P$ at time $T = \tilde{O}(L)$[1]. We show case this difference by implementing time-independent Glauber dynamics via a bi-directional transformer based masked language models. We describe this procedure based on (Wang & Cho, 2019) in Appendix F. Our experiments (see Table 1, BERT-large w/ Gibbs sampling) show that even with a large number of steps, the quality of generation with this method is much worse than GGM (see Figure 2a).

## 3 GLAUBER GENERATIVE MODEL (GGM)

Diffusion models in the continuous domain contain 3 crucial steps: (i) forward process: where a sample from the target distribution is gradually noised to obtain a sample from the standard Gaussian distribution over many iterations, (ii) reverse process: which time reverses the forward process – i.e., it denoises a Gaussian random vector into a sample from the target distribution, and (iii) model training: where the reverse process is learned with data from the forward process. We follow the same recipe to describe GGM below. Consider a finite set $\mathcal{X}$ and $L \in \mathbb{N}$. Suppose $P^*$ is the target over $\mathcal{X}^L$ (i.e., sequences of length $L$ from $\mathcal{X}$).

### 3.1 FORWARD PROCESS

Fix an element $\phi \notin \mathcal{X}$. At time $t \in \{0, \ldots, T-1\}$, let $\Pi_t$ denote a distribution over $\mathcal{X} \cup \{\phi\}$. We think of $\Pi_t$ as the noise distribution. Let $i_0, \ldots, i_{T-1} \in \{0, \ldots, L-1\}$ be fixed. We pick $i_0, \ldots, i_{T-1}$ in a round robin way with respect to some permutation in order to ensure that each coordinate appears in $i_0, \ldots, i_{T-1}$ at-least $T/L$ times. Suppose $X_0$ is a sample from $P^*$. At time $t$, the coordinate $i_t$ of $X_t$ is noised using $\Pi_t$ in order to obtain $X_{t+1}$: Draw a token $Z_t \sim \Pi_t$ independent of $X_t$. Set:

$$X_{t+1,j} = \begin{cases} X_{t,j} \text{ if } j \neq i_t \\ X_{t,j} \text{ if } j = i_t \text{ and } Z_t = \phi \\ Z_t \text{ if } j = i_t \text{ and } Z_t \in \mathcal{X} \end{cases} \tag{1}$$

That is, $X_{t+1} = X_t$ with probability $\Pi_t(\phi)$ and $X_{t+1,i_t}$ is sampled from $\Pi_t(\cdot|\mathcal{X})$ with probability $1 - \Pi_t(\phi)$. It is easy to obtain $X_t, X_{t+1}$ directly from $X_0$ as described in Appendix H. We denote the distribution of $X_t$ by $P_t$.

**Lemma 1.** *Suppose that $\Pi_t(\cdot|\mathcal{X}) = \Pi(\cdot|\mathcal{X})$ is the same for every $t$. Suppose $\Pi_t(\phi) \leq 1 - \epsilon$ for some $\epsilon > 0$. As $T \to \infty$, the distribution of $X_T$ converges to $\Pi(\cdot|\mathcal{X})^{\otimes L}$ in total variation distance. Specifically, we have:*

$$\mathsf{TV}(\mathsf{Law}(X_T), \Pi(\cdot|\mathcal{X})^{\otimes L}) \leq L(1-\epsilon)^{\lfloor T/L \rfloor}$$

The proof of this lemma is given in Appendix A. Under the choice $\Pi_t(\cdot|\mathcal{X}) = \mathsf{Unif}(\mathcal{X})$ and $\Pi_t(\phi) = \frac{1}{2}$. In this case, Lemma 1 shows that the forward process converges to $\mathsf{Unif}(\mathcal{X})^L$ – i.e., each token is chosen i.i.d uniformly at random from $\mathcal{X}$.

---

[1]In the rest of the paper, our Glauber dynamics goes reverse in time: $X_{t-1,i} \sim P_t(X_t|X_{t,-i})$ as is the convention in the diffusion model literature.

## 3.2 REVERSE PROCESS VIA GLAUBER DYNAMICS

Consider $P_t$ as given in the forward process. Suppose we have $\hat{X}_{t+1} \sim P_{t+1}$. Then, the reverse process seeks to obtain a sample $\hat{X}_t \sim P_t$. We call the following time dependent Glauber dynamics update to sample $X_t$ as the reverse Glauber dynamics:

1. $\hat{X}_{t,j} = \hat{X}_{t+1,j}$ for all $j \neq i_t$
2. Sample $\hat{X}_{t,i_t}$ from the probability distribution $\mathbb{P}(X_{t,i_t} = \cdot | X_{t+1,-i_t} = \hat{X}_{t,-i_t})$

**Lemma 2.** *Suppose $\hat{X}_T \sim P_T$. Suppose $\hat{X}_0$ is obtained by applying the reverse Glauber Dynamics $T$ times as defined above. Then, $\hat{X}_0 \sim P^*$.*

*Proof.* Let $X_0, \ldots, X_T$ denote the forward process described in Section 3.1. We prove via induction that for any $t \in \{0, \ldots, T-1\}$, $\hat{X}_{t+1} \sim P_{t+1}$ implies $\hat{X}_t \sim P_t$. Indeed, for any $x \in \mathcal{X}^L$, consider:

$$
\begin{aligned}
\mathbb{P}(\hat{X}_t = x) &= \mathbb{P}(\hat{X}_{t+1,-i_t} = x_{-i_t})\mathbb{P}(X_{t,i_t} = x_{i_t}|X_{t+1,-i_t} = x_{-i_t}) \\
&= \mathbb{P}(X_{t+1,-i_t} = x_{-i_t})\mathbb{P}(X_{t,i_t} = x_{i_t}|X_{t+1,-i_t} = x_{-i_t}) = \mathbb{P}(X_t = x) = P_t(x)
\end{aligned}
\tag{2}
$$

The first step follows from the definition of the reverse Glauber Dynamics. The second step follows from the induction hypothesis. Thus we conclude the result. □

**Remark 1.** *The forward process introduced in D3PM (Austin et al., 2021) noises all tokens at once and considers a general class of noising processes. We consider a specific class of noising processes, and apply it to one token at a time. As we show below, this allows us to implement the reverse process by predicting single tokens instead of predicting for all tokens like in (Austin et al., 2021; Gu et al., 2021). Moreover, this approach results in an exact reduction to binary classification. Prior approaches to Discrete Diffusion Models had to learn the transition probability of token value $a \in \mathcal{X}$ to token value $b \in \mathcal{X}$ at position $i$, scaling quadratically with $|\mathcal{X}|$ sized transition matrix resulting in suboptimal performance (Hu et al., 2021). Our training approach described below learns the probability that token $a \in \mathcal{X}$ appearing at position $i$ in $X_{t+1}$ was present in position $i$ of $X_t$ or not, which scales linearly in $|\mathcal{X}|$.*

## 3.3 MODEL TRAINING

We now describe how to train a neural network with data from the forward process (Section 3.1) in order to implement the reverse Glauber dynamics (Section 3.2). The following lemma is the key to our learning algorithm. We refer to Appendix B for its proof.

**Lemma 3.** *Suppose $x \in \mathcal{X}^n$. Suppose that $(X_t)_{t=0,..,T-1}$ denotes the forward process. Then, for any $a \in \mathcal{X}$:*

$$
\mathbb{P}(X_{t,i_t} = a|X_{t+1,-i_t} = x_{-i_t}) = \frac{\mathbb{P}(Z_t = a)}{\mathbb{P}(Z_t = \phi)} \left( \frac{1}{\mathbb{P}(Z_t = a|X_{t+1,-i_t} = x_{-i_t}, X_{t+1,i_t} = a)} - 1 \right)
\tag{3}
$$

Therefore, we can estimate $\mathbb{P}(X_{t,i_t} = a|X_{t+1,-i_t} = x_{-i_t})$ using an estimate for $\mathbb{P}(Z_t = x_{i_t}|X_{t+1} = x)$. The latter is equivalent to learning $\mathbb{P}(Z_t = a|X_{t+1,-i_t} = x_{-i_t}, X_{t+1,i_t} = a)$ for every $a \in \mathcal{X}$.

**Reduction to Binary Classification:** Given $a \in \mathcal{X}$ and time $t$, we consider the distribution $\mathcal{D}_{t,a} = \text{Law}((X_{t+1,-i_t}, \mathbb{1}_{Z_t=a}|X_{t+1,i_t} = a))$. Given a sample $X_0 \sim P^*$, we can obtain $X_0, X_t, Z_t, X_{t+1}$ according to the forward process, allowing us to sample from $\mathcal{D}_{t,a}$. Consider the following binary classification problem (denoted by $\mathcal{B}_t(P^*, \Pi, a)$) corresponding to the data distribution $\mathcal{D}_{t,a}$:

$$\text{Predict } \mathbb{1}_{Z_t=a} \text{ given } X_{t+1,-i_t}$$

That is, we try to predict if $X_{t+1,i} = a$ because of pure noise (i.e., $Z_t = a$) or if it is because $X_{t,i} = a$ (signal). This can be solved by minimizing the cross entropy loss over a model class with input $X_{t+1,-i_t}$. Such a model learns to predict $\mathbb{P}(Z_t = a|X_{t+1,-i_t} = x_{-i_t}, X_{t+1,i_t} = a)$.

**Training Neural Network:** Consider a mask token $\omega \notin \mathcal{X}$, and parameterize a neural network $f_\theta : (\mathcal{X} \cup \{\omega\})^L \times \{0, \ldots, T-1\} \to [0,1]^{\mathcal{X}}$ to solve the sequence of binary classification problems $\mathcal{B}_t(P^*, \Pi, a)$ for $t \in \{0, \ldots, T-1\}, a \in \mathcal{X}$. The neural network input is $(x', t)$ where $x' \in$

$(\mathcal{X} \cup \{\omega\})^L$, $x'_{t+1,j} \in \mathcal{X}$ for all $j \neq i_t$ and $x'_{i_t} = \omega$. The output logit corresponding to $a \in \mathcal{X}$ solves $\mathcal{B}_t(P^*, \Pi, a)$. That is, $f_\theta$ outputs $\hat{y} = f_\theta(x', t) \in [0, 1]^{\mathcal{X}}$ where $\hat{y}_a$ models $\mathbb{P}(Z_t = a | X_{t+1,-i_t} = x'_{-i_t}, X_{t+1,i_t} = a)$. Note that using a special token like $\omega$ allows us to compute $\hat{y}_a$ for all $a \in \mathcal{X}$ in parallel in a single forward pass using the standard transformer architecture. The loss function and the training algorithm is given in Algorithm 1.

---

**Algorithm 1:** Training a Glauber Generative Model (GGM)

---

**Input:** Dataset $\mathcal{D}$, timesteps $T$, optimizer `opt`, model $f$, positions $\{i_t\}_{t=1}^T$, $\{\Pi_t\}_{t=1}^T$
**Result:** Trained parameters $\theta$

1 Initialize $\theta$, initialize the optimizer state `opt.initialize`$(\theta)$;
2 **for** *each iteration* **do**
3     Sample $X_0 \sim \mathcal{D}$, $t \sim \mathsf{Unif}(\{0, \ldots, T-2\})$;
4     Get $X_t, Z_t, X_{t+1} \leftarrow$ `forward_process`$(X_0, t, \{i_s\}_{s=1}^t, \{\Pi_s\}_{s=1}^t)$ // more
        details about the forward process are given in Appendix H
5     Set $\hat{X}'_{t+1,j} \leftarrow X_{t+1,j}$ for all $j \neq i_t$, $\hat{X}'_{t+1,i_t} \leftarrow \omega$;      // $\omega$ is the mask token
6     Compute loss
        $\mathcal{L}(\theta; Z_t, X'_{t+1}) = -\mathbb{1}_{Z_t \neq \phi} \log \left(f_\theta(X'_{t+1}, t)_{x_{t+1,i_t}}\right) - \mathbb{1}_{Z_t = \phi} \log \left(1 - f_\theta(X'_{t+1}, t)_{x_{t+1,i_t}}\right)$;
7     $\theta \leftarrow$ `opt.update`$(\theta, \nabla_\theta \mathcal{L}(\theta; Z_t, X_{t+1}))$;      // any optimizer like AdamW,
        etc.
8 **end**
9 **return** $\theta$;

---

**Implementing the Reverse Process:** We now use the trained neural network to approximate the reverse Glauber dynamics. By Lemma 1, whenever $T$ is large $P_T \approx \Pi(\cdot|\mathcal{X})^L$. We sample $\hat{X}_T \sim \Pi(\cdot|\mathcal{X})^L$, the pure noise distribution. Note that it is now sufficient to estimate $\hat{\mathbb{P}}(X_{t,i_t} = a | X_{t+1,-i_t})$. Given $\hat{X}_{t+1}$, let $\hat{X}'_{t+1}$ be its masked version in the position $i_t$. Let $\hat{y} = f_\theta(\hat{X}'_{t+1}, t)$. By Lemma 3, we can estimate $\mathbb{P}(X_{t,i_t} = a | X_{t+1,-i_t} = \hat{X}_{t,-i_t})$ in step 2 of the reverse process with: $\hat{\mathbb{P}}(X_{t,i_t} = a | X_{t+1,-i_t}) = \frac{\Pi_t(a)}{\Pi_t(\phi)} \left(\frac{1}{\hat{y}_a} - 1\right)$.

---

**Algorithm 2:** Inference from a Glauber Generative Model (GGM)

---

**Input:** Timesteps $T$, trained parameters $\theta$, model $f$, noise distributions $\{\Pi_t\}_{t=0}^T$
**Result:** Sample $\hat{X}_0$ from the target distribution $\Pi^*$

1 Initialize $\hat{X}_T \sim \Pi_T(\cdot|\mathcal{X})^L$;
2 **for** $t \leftarrow T-1$ **to** $0$ **do**
3     Set $\hat{X}'_{t+1,j} \leftarrow X_{t+1,j}$ for all $j \neq i_t$, $\hat{X}'_{t+1,i_t} \leftarrow \omega$;      // $\omega$ is the mask token
4     Get $\hat{y} = f_\theta(\hat{X}'_{t+1}, t)$;
5     Compute $\hat{\mathbb{P}}(X_{t,i_t} = a | X_{t+1,-i_t}) = \frac{\Pi_t(a)}{\Pi_t(\phi)} \left(\frac{1}{\hat{y}_a} - 1\right)$ for all $a$;
6     Set $\hat{X}_{t,j} \leftarrow \hat{X}_{t+1,j}$ for all $j \neq i$;
7     Sample $\hat{X}_{t,i_t} \sim \hat{\mathbb{P}}(\cdot | X_{t+1,-i_t})$;      // can do top-$p$ sampling, etc. here
8 **end**
9 **return** $\hat{X}_0$;

---

### 3.4 CONVERGENCE OF GGM

We provide a bound on the total variation distance (TV) between the output distribution $\hat{P}_0$ of Algorithm 2 and the target distribution $P^*$ in Theorem 1. We provide the proof in Appendix C.

**Theorem 1.** *Suppose that* $\Pi_t = \Pi$ *for every* $t$ *and* $\Pi(\phi) = 1 - p$ *for some* $p \in (0, 1)$. *Suppose that our model learns* $\mathbb{P}(Z_t = a | X_{t+1,-i} = x_{-i}, X_{t+1,i} = a)$ *perfectly from Algorithm 1. If* $T \geq L \frac{\log(L/\delta)}{\log(1/1-p)}$ *then the output distribution* $\hat{P}_0$ *of Algorithm 2 satisifes:*

$$\mathsf{TV}(\hat{P}_0, P^*) \leq \delta \tag{4}$$

### 3.5 CONDITIONAL INFERENCE

While Algorithm 2 gives unconditional generations from a target distribution $P^*$, we now describe a simple modification to obtain conditional generation without any additional training. Suppose the tokens at places $J = (j_1, \ldots, j_K)$ are conditioned to be $(c_1, \ldots, c_K)$. Then define $\hat{X}_T$ such that $\hat{X}_{T,i} \sim \Pi_T(\cdot | \mathcal{X})$ for $i \notin J$ and $\hat{X}_{T,j_k} = c_k$ for all $k = 1, \ldots, K$. We run Algorithm 2 as before, but do not update the tokens in the positions given by $J$. Using this, our models can generate tokens under a variety of different zero-shot control settings, like prefix/suffix completion, arbitrary infilling, and arbitrary lexical constraints. For example, we provide examples of conditional generations for language in Appendix L.1, where our models infill the middle $512$ tokens given the first and last $256$ tokens. Figure 4b shows examples of conditional generations from our model on CelebA-HQ where our model is able to infill a variety of different pixel-space masks. We provide more details regarding obtaining the prompt tokens and positions from the pixel masks in Appendix D.

### 3.6 ARCHITECTURE

Following (Lou et al., 2023) closely, we design $f_\theta$ to be a transformer model, based on the DiT model family (Peebles & Xie, 2022). We find that encoding time using the `adaLN-Zero` approach used in (Peebles & Xie, 2022) improves the convergence time of our models significantly. Like (Lou et al., 2023) and (Gulrajani & Hashimoto, 2023), we too use rotary embeddings (Su et al., 2021) to encode position. When the $i^{\text{th}}$ token to the model is masked (using $\omega$), we pass the final layer representation for the $i^{\text{th}}$ token to the classifier that returns $\hat{y} \in [0,1]^{|\mathcal{X}|}$ after applying the sigmoid activation on the logits. We provide hyperparameters and more architectural details in Appendix I.

## 4 RESULTS

### 4.1 LANGUAGE GENERATION

We train our models on the OpenWebText dataset (Gokaslan & Cohen, 2019) and evaluate on language generation. We compare with recent diffusion-based models for language (Plaid (Gulrajani & Hashimoto, 2023), SEDD-medium (Lou et al., 2023), and MDLM (Sahoo et al., 2024)) and GPT2 models of different sizes in Table 1. Following (Lou et al., 2023; Han et al., 2022; Dieleman et al., 2022), we evaluate unconditional generations of length $1024$ tokens using generative perplexity (Gen. PPL) measured by a different, larger, autoregressive model like GPT-Neo-2.7B (Black et al., 2021). Our model, with $387M$ parameters, outperforms SEDD-medium ($424M$ parameters) and MDLM, the previous state-of-the-art discrete diffusion models. Our model performs competitively with Plaid (1.3B parameters) with $3.4\times$ less parameters and after being trained on a smaller dataset (OpenWebText instead of OpenWebText2 (Gao et al., 2020)). However, all diffusion-based approaches lie significantly behind GPT2 models of comparable sizes. Following (Wang & Cho, 2019), we include a MLM-based baseline which implements time independent Glauber dynamics as described in Section 2.2 to compare to time dependent Glauber dynamics of GGM. We begin with a noisy sequence of tokens, and take multiple token-by-token passes at the token sequence. At each position, we mask the corresponding token and pass the resulting sequence to a cased BERT-large model ($334M$ parameters) (Devlin et al., 2019) to re-sample token value in this position. We generate and evaluate sequences of length $512$ for BERT (maximum length that BERT allows). This approach is able to generate short phrases but is unable to generate text that remains coherent over $512$ tokens and gets poor perplexity. We provide the exact sampling algorithm in Appendix F. When sampling from GPT2 models, we force the models to generate text of length $1024$ by setting the probability of the end-of-text token to $0$. For Plaid, SEDD-medium, and MDLM, we use the default sampling algorithms as used by the authors in (Gulrajani & Hashimoto, 2023), (Lou et al., 2023), and (Sahoo et al., 2024) respectively. We include example generations from our model in Appendix L. Hyperparameters and other training details are provided in Appendix I.

**Remark 2.** *During sampling, we use top-$p$ sampling, whereas SEDD and MDLM do not. However, recent work (Zheng et al., 2024, Section 5) shows that when using the Gumbel-max trick for sampling from a categorical distribution (as used by SEDD and MDLM, but not by our method), using FP32 precision instead of FP64 behaves equivalently to lowering the temperature during sampling, as a result of truncation. When this is fixed using FP64 precision, the default samplers for SEDD and MDLM result in a generative perplexity of about $100$ (more than $3\times$ the reported values).*

Table 1: Generative Perplexities evaluated over 1024 unconditional generations. We do not evaluate a larger model (e.g., GPT2-xl) with a smaller model (e.g., GPT2-small).

| | Evaluation Model | | GPT2-large (774**M**) | GPT2-xl (1.6**B**) | GPT-neo (2.7**B**) |
|---|---|---|---|---|---|
| **Evaluated Model** | **Sampling Algorithm** | **Total Params** | **Gen. PPL (↓)** | **Gen. PPL (↓)** | **Gen. PPL (↓)** |
| **Autoregressive Models** | | | | | |
| GPT2-medium (Radford et al., 2019) | top-$p$, $p = 0.8$ $L = 1024, T = 1024$ | 345M | 12.4 | 13.0 | 14.5 |
| GPT2-large (Radford et al., 2019) | top-$p$, $p = 0.8$ $L = 1024, T = 1024$ | 774M | – | 6.5 | 7.4 |
| GPT2-xl (Radford et al., 2019) | top-$p$, $p = 0.8$ $L = 1024, T = 1024$ | 1.6B | – | – | 6.8 |
| **Masked Language Models** | | | | | |
| BERT-large w/ Gibbs sampling (Wang & Cho, 2019) | top-$p$, $p = 0.8$ $L = 512, T = 2048$ | 334M | 487.0 | 487.5 | 488.7 |
| BERT-large w/ Gibbs sampling (Wang & Cho, 2019) | top-$p$, $p = 0.8$ $L = 512, T = 65536$ | 334M | 28.7 | 28.4 | 26.4 |
| **Diffusion Models** | | | | | |
| Plaid (Gulrajani & Hashimoto, 2023) | $\tau = 0.9$ as per (Gulrajani & Hashimoto, 2023) $L = 1024, T = 4096$ | 1.3B | 19.7 | 19.7 | 17.9 |
| SEDD-medium (Lou et al., 2023) | default as per (Lou et al., 2023) $L = 1024, T = 2048$ | 424M | 27.3 | 28.0 | 25.2 |
| MDLM (Sahoo et al., 2024) | default as per (Sahoo et al., 2024) $L = 1024, T = 1000$ | 170M | 44.2 | 45.4 | 40.9 |
| **GGM (ours)** | top-$p$, $p = 0.8$ $L = 1024, T = 4096$ | 387M | 19.5 | 19.9 | 18.0 |

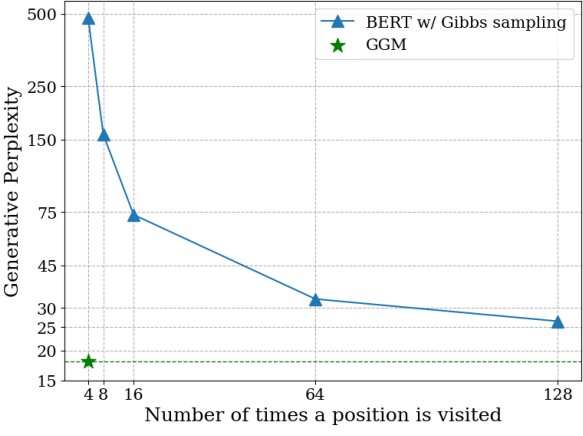

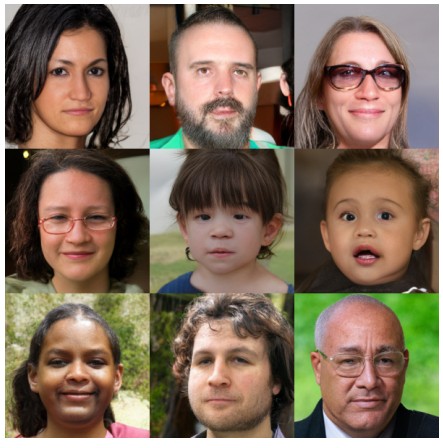

(a) Generative perplexity of GGM vs BERT-large with Gibbs sampling. Inference is run for $T = KL$ timesteps, where $K$ is the number of visits per position (plotted on the $x$-axis).

(b) Examples of $256{\times}256$ unconditional generations from our model on FFHQ. We show more examples in Appendix G.

## 4.2  IMAGE GENERATION

We tokenize images and de-tokenize tokens using an off-the-shelf VQGAN-based tokenizer used by the MUSE model (Chang et al., 2023). The tokenizer is trained on a large, general image corpus with a downsampling ratio $f = 8$. Thus, a $256 \times 256$ corresponds to $32 \times 32 = 1024$ tokens. We report the FID (Parmar et al., 2022) values in Table 2 for unconditional $256{\times}256$ image synthesis on the CelebA-HQ dataset (Karras et al., 2017) and the FFHQ dataset (Karras et al., 2018). On CelebA-HQ, our model outperforms several autoencoder-based baselines by a large margin and beats a much larger 801M parameter GPT2-based autoregressive model that also works with VQGAN tokens (Esser et al., 2021). We also outperform VQ-DDM (Hu et al., 2021), another model that applies discrete diffusion similar to (Hoogeboom et al., 2021) in the latent space of a VQGAN. On FFHQ, our method outperforms the autoencoder baselines, the baseline version of StyleSwin (Zhang et al.,

2021), the PixelSNAIL (Chen et al., 2017) model on VQGAN tokens (Esser et al., 2021), and is competitive with other kinds of models like BigGAN (Brock et al., 2018).

**Comparison with SoTA methods:** We find a substantial gap in FID between our approach and SoTA methods like LDMs (Rombach et al., 2021), VQ-Diffusion (Gu et al., 2021), and StyleSwin (Zhang et al., 2021). Based on the findings in MAGVIT-v2 (Yu et al., 2024), we believe that this gap can be brought down significantly, if not completely eliminated, by the use of more expressive and dataset-specific tokenizers. This can bring down the required sequence length, lead to more expressive and relevant tokens in the vocabulary, and thereby ease downstream sequence modeling. For example, the dataset-specific tokenizers used in (Esser et al., 2021) and (Gu et al., 2021)) require only 256 tokens for CelebA-HQ and FFHQ, while the general-purpose tokenizer used by our model leads to 1024 tokens. StyleSwin (Zhang et al., 2021) uses several techniques to boost generation quality. More specifically, the baseline StyleSwin model gets an FID of 15.0 on FFHQ (in comparison, our method achieves 12.5). However, with techniques such as style injection, double attention, wavelet discriminator, sinusoidal positional encoding at each generation scale, and balanced consistency regularization, StyleSwin is able to achieve an FID of 2.8 (Table 5 of (Zhang et al., 2021)). We show unconditionally generated examples from our model trained on CelebA-HQ in Figure 4a and from our model trained on FFHQ in Figure 2b and Appendix G. In Appendix E, we show the nearest neighbors for our generations from the CelebA-HQ training data. In Figure 3a, we visualize the forward noising process applied on an real image. In Figure 3b, we visualize the reverse denoising process leading to a generation.

Table 2: FID scores computed using 50K $256 \times 256$ unconditionally generated images on CelebA-HQ and FFHQ. Baseline results are replicated from (Esser et al., 2021; Jiang et al., 2021; Vahdat et al., 2021; Hu et al., 2021; Jiang et al., 2021; Gu et al., 2021). ‡ denotes that the model uses a dataset-specific tokenizer / latent space. We provide more details in Appendix I.

| Model Family | CelebA-HQ | | FFHQ | |
|---|---|---|---|---|
| | **Model** | **FID ($\downarrow$)** | **Model** | **FID ($\downarrow$)** |
| **Flow & Autoencoder** | GLOW (Kingma & Dhariwal, 2018) | 69.0 | VDVAE ($\tau = 0.8$) (Child, 2020) | 29.8 |
| | Style ALAE (Pidhorskyi et al., 2020) | 19.2 | VDVAE ($\tau = 1.0$) (Child, 2020) | 33.5 |
| | DC-VAE (Parmar et al., 2020) | 15.8 | VDVAE ($\tau = 0.9$) (Child, 2020) | 28.5 |
| **GAN** | TransGAN (Jiang et al., 2021) | 10.3 | BigGAN (Brock et al., 2018) | 12.4 |
| | PGGAN (Karras et al., 2017) | 8.0 | StyleSwin (baseline) (Zhang et al., 2021) | 15.0 |
| | StyleSwin (Zhang et al., 2021) | 3.3 | StyleSwin (Zhang et al., 2021) | 2.8 |
| **Continuous Diffusion** | ‡LSGM (Vahdat et al., 2021) | 7.2 | VE (Jolicoeur-Martineau et al., 2021) | 15.7 |
| | ‡LDM-4 (Rombach et al., 2021) | 5.1 | ‡LDM-4 (Rombach et al., 2021) | 5.0 |
| **Autoregressive** | ‡VQGAN + GPT2 (Esser et al., 2021) | 10.2 | ‡VQGAN + PixelSNAIL (Chen et al., 2017) | 21.9 |
| | | | ‡VQGAN + GPT2 (Esser et al., 2021) | 9.6 |
| **Discrete Diffusion** | ‡VQ-DDM (w/o ReFiT) (Hu et al., 2021) | 22.6 | ‡VQ-Diffusion (Gu et al., 2021) | 6.3 |
| | ‡VQ-DDM (w/ ReFiT) (Hu et al., 2021) | 13.2 | | |
| | MUSE tokens (Chang et al., 2023) + GGM (**ours**) | 9.8 | MUSE tokens (Chang et al., 2023) + GGM (**ours**) | 12.5 |

## 5 DISCUSSION AND LIMITATIONS

We propose a novel conceptual framework for discrete diffusion and evaluate its performance on generative modeling tasks for language and images. We show that our proposed approach outperforms existing discrete diffusion models for language generation and demonstrates strong performance for image generation without using dataset-specific image tokenizers. Our approach is theoretically principled and provides a promising new direction for multi-modal generative modeling. Future works can explore various applications and extensions like protein modeling (Meshchaninov et al., 2024; Campbell et al., 2024) and generalist agents (Reed et al., 2022). In its current state, we find that discrete diffusion models lag behind state of the art methods like autoregressive LLMs for language, and GANs and continuous diffusion models for image generation. We note that state of the art image generation algorithms are a product of years of cutting edge research, and utilize numerous additional customized techniques (such as data-specific tokenizers, style injection, etc.) that

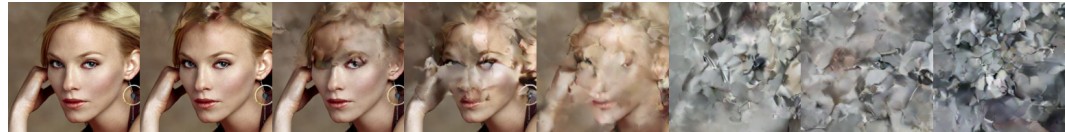

(a) Noising process at different timesteps visualized using the MUSE tokenizer. From left to right, the timesteps are $t = 0, 256, 512, 768, 1024, 2048, 3072, 4095$.

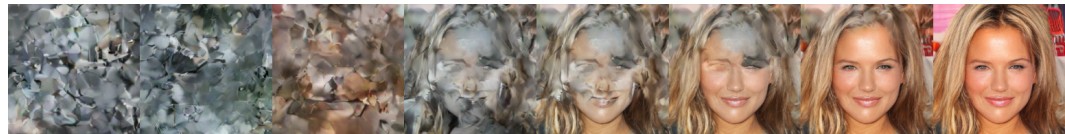

(b) Denoising process at different times visualized using the MUSE tokenizer. From left to right, the timesteps are $t = 4095, 3072, 2048, 1024, 768, 512, 256, 0$.

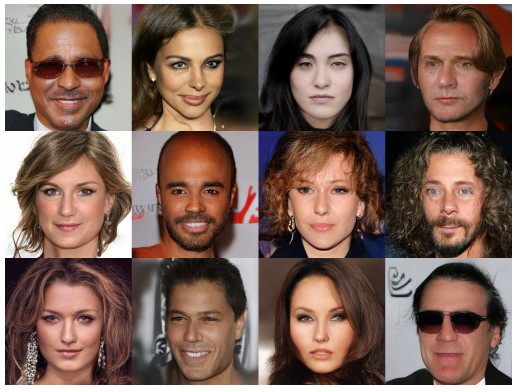

(a) Examples of $256 \times 256$ unconditional generations from our model on CelebA-HQ.

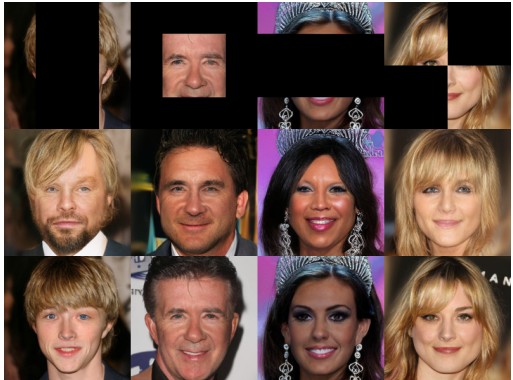

(b) Comparison of $256 \times 256$ conditional generations (middle) from our model on CelebA-HQ given masked inputs (top) with the ground-truth images (bottom).

can improve performance dramatically (for example, Table 5 of (Zhang et al., 2021)). We believe that with further research and refinement, discrete diffusion models can obtain even better results.

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

## A  PROOF OF LEMMA 1

*Proof.* We consider Proposition 4.7 in Levin & Peres (2017), which establishes a coupling characterization for the total variation distance. This implies that for any random variable $Y \sim \Pi(\cdot|\mathcal{X})^{\otimes L}$ which is jointly distributed with $X_0, \ldots, X_T$, we must have:

$$\mathsf{TV}(\mathsf{Law}(X_T), \Pi(\cdot|\mathcal{X})^{\otimes L}) \leq \mathbb{P}(X_T \neq Y) \tag{5}$$

We construct such a joint distribution/ coupling below. Let $X_0, \ldots, X_T$ be obtained using the forward process. Consider the event $E_{T,i} := \{\exists t : 0 \leq t \leq T-1, Z_t \neq \phi \text{ and } i_t = i\}$ and the event $E_T = \cap_{i=0}^{L-1} E_{T,i}$. This means that each coordinate of $X_0$ has been noised at-least once in time $T$. Define $\tau(i) = \sup\{t : 0 \leq t \leq T-1, Z_t \neq \phi \text{ and } i_t = i\}$. We define the random variable $Y$ as follows:

$$Y_i = \{Z_{\tau(i)} \text{ under the event } E_{T,i} \sim \Pi(\cdot|\mathcal{X}) \text{ independently otherwise} \tag{6}$$

It is easy to show that $Y \sim \Pi(\cdot|\mathcal{X})^{\otimes L}$. By definition of $Y$, $\{Y \neq X_T\} \subseteq E_T^{\mathsf{C}}$. Therefore, we conclude from Equation equation 5 that:

$$\mathsf{TV}(\mathsf{Law}(X_T), \Pi(\cdot|\mathcal{X})^{\otimes L}) \leq \mathbb{P}(E_T^{\mathsf{C}}) \leq L(1-\epsilon)^{\lfloor T/L \rfloor} \overset{T \to \infty}{\to} 0$$

$\square$

## B  PROOF OF LEMMA 3

*Proof.* Note the following relationships between events

1. $\{X_{t+1} = x\} \cap \{Z_t = \phi\} = \{Z_t = \phi\} \cap \{X_t = x\}$
2. For $z \neq \phi$ and $x \in \mathcal{X}^L$, we have:

$$\{X_{t+1} = x\} \cap \{Z_t = z\} = \begin{cases} \{X_{t,-i_t} = x_{-i_t}\} \cap \{Z_t = z\} & \text{if } x_{i_t} = z \\ \emptyset & \text{otherwise} \end{cases}$$

3. For any $x \in \mathcal{X}^L$,

$$\{X_{t+1} = x\} = (\{X_t = x\} \cap \{Z_t = \phi\}) \cup (\{X_{t,-i_t} = x_{-i_t}\} \cap \{Z_t = x_{i_t}\})$$

From the relationships above, we have:

$$\begin{aligned} \mathbb{E}[\mathbb{1}_{Z_t=z}|X_{t+1} = x] &= \mathbb{P}(\mathbb{1}_{Z_t=z} = 1|X_{t+1} = x) \\ &= \frac{\mathbb{P}(Z_t = z, X_{t+1} = x)}{\mathbb{P}(X_{t+1} = x)} \\ &= \begin{cases} 0 & \text{if } z \neq \phi \text{ and } z \neq x_{i_t} \\ \frac{\mathbb{P}(Z_t=\phi)\mathbb{P}(X_t=x)}{\mathbb{P}(X_{t+1}=x)} & \text{if } z = \phi \\ \frac{\mathbb{P}(Z_t=x_{i_t})\mathbb{P}(X_{t,-i_t}=x_{-i_t})}{\mathbb{P}(X_{t+1}=x)} & \text{if } z = x_{i_t} \end{cases} \end{aligned} \tag{7}$$
$$\mathbb{P}(X_{t+1} = x) = \mathbb{P}(Z_t = \phi)\mathbb{P}(X_t = x) + \mathbb{P}(Z_t = x_{i_t})\mathbb{P}(X_{t,-i_t} = x_{-i_t})$$

Combining these relationships, we have:

$$\begin{aligned} \mathbb{P}(Z_t = x_{i_t}|X_{t+1} = x) &= \frac{1}{1 + \frac{\mathbb{P}(Z_t=\phi)}{\mathbb{P}(Z_t=x_{i_t})}\mathbb{P}(X_{t,i_t} = x_{i_t}|X_{t,-i_t} = x_{-i_t})} \\ \implies \mathbb{P}(X_{t,i_t} = x_{i_t}|X_{t,-i_t} = x_{-i_t}) &= \frac{\mathbb{P}(Z_t = x_{i_t})}{\mathbb{P}(Z_t = \phi)}\left(\frac{1}{\mathbb{P}(Z_t = x_{i_t}|X_{t+1} = x)} - 1\right) \end{aligned} \tag{8}$$

$\square$

## C   PROOF OF THEOREM 1

*Proof.* Suppose we initialize $\hat{X}_T \sim P_T$ in Algorithm 2 instead of $\Pi_T(\cdot|\mathcal{X})^L$. By Lemma 2 and Lemma 3 we conclude that if the model learns $\mathbb{P}(Z_t = a|X_{t+1,-i} = x_{-i}, X_{t+1,i} = a)$ perfectly, then the output of Algorithm 2 satisfies $\hat{X}_0 \sim P^*$. That is, it yields a sample from the exact target distribution. However, we do not initialize with $\hat{X}_T \sim P_T$, but with $\hat{X}_T \sim \Pi_T(\cdot|\mathcal{X})^L$ and this yields an output $\hat{X}_0 \sim \hat{P}_0 \neq P^*$.

Notice that Algorithm 2 is a Markov Process whose output distribution is $P^*$ if the input distribution is $P_T$ and the output distribution is $\hat{P}_0$ if the input distribution if $\Pi(\cdot|\mathcal{X})^L$. Therefore we can write down the data processing inequality in this case as:

$$
\begin{aligned}
\mathsf{TV}(\hat{P}_0, P^*) &\leq \mathsf{TV}(P_T, \Pi(\cdot|\mathcal{X})^L) \\
&\leq L(1-p)^{-\lfloor T/L \rfloor}
\end{aligned}
\tag{9}
$$

In the second step, we have applied Lemma 1. Setting the RHS above $\leq \delta$, we conclude the result.

$\square$

# D ZERO-SHOT CONDITIONAL GENERATION WITH GGM

For image data, we can mask/corrupt the inputs either in the token space or in the original pixel space.

## D.1 MASKING IN THE TOKEN SPACE

Given a tokenized image $X_0 \sim P^*$, we generate masked images by setting $X_{0,a:b} \sim \Pi(\cdot|\mathcal{X})$ and run the conditional inference as specified in Algorithm A2. Here, $a : b$ denotes the slice of tokens from index $a$ to index $b$. In Figure 5 we show examples with $a = 256$ and $b = 768$ (that is, $X_{0,:256}$ and $X_{0,768:}$ form our prompt tokens $X^*$ and the prompt positions $J = \{0, \ldots, 255, 768, \ldots, 1023\}$).

## D.2 MASKING IN THE PIXEL SPACE

Instead of masking in the token space, we can also mask in the original pixel space, pre-tokenization. In this case, we simulate the previous setting using the available pixel mask. Given an image $Y \in [0, 1]^{H \times W}$ and a mask $M \in \{0, 1\}^{H \times W}$, we generate two versions of the masked image $Y_1$ with masked pixels set to 0 and $Y_2$ with the masked pixels set to 1. We then tokenize these two versions and compare the tokens to get the mask in the token space. That is, if $X_1, X_2 \in \mathcal{X}^L$ are obtained after tokenizing $Y_1, Y_2$, we set $J = \{j \in [L] : X_{1,j} = X_{2,j}\}$ and run Algorithm A2 with this $J$ and prompt tokens $\{X_j^* = X_{1,i} = X_{2,j}\}_{j \in J}$. We show results of this approach for a variety of pixel masks in Figure 4b and Figure 6.

---

**Algorithm A2:** Conditional Inference from a Glauber Generative Model (GGM)

**Input:** Timesteps $T$, trained parameters $\theta$, model $f$, noise distribution $\Pi_t$, prompt positions $J$, prompt tokens $\{X_j^*\}_{j \in J}$

**Result:** Sample $\hat{X}_0$ from the target distribution $\Pi^*$ with $\hat{X}_{0,j} = X_j^*$ for all $j \in J$

1   Initialize $\hat{X}_T \sim \Pi_T(\cdot|\mathcal{X})^L$, fix $\hat{X}_{T,j} = X_j^*$ for all $j \in J$;

2   **for** $t \leftarrow T - 1$ **to** 0 **do**

3     **if** $i_t \in J$ **then**

4       Set $\hat{X}_{t,i_t} \leftarrow X_{i_t}^*$;

5     **else**

6       Set $\hat{X}'_{t+1,j} \leftarrow X_{t+1,j}$ for all $j \neq i_t$, $\hat{X}'_{t+1,i_t} \leftarrow \omega$;

7       Get $\hat{y} = f_\theta(\hat{X}'_{t+1}, t)$;

8       Compute $\hat{\mathbb{P}}(X_{t,i_t} = a|X_{t+1,-i_t}) = \frac{\Pi_t(\hat{X}_{t+1,i_t})}{\Pi_t(\phi)} \left( \frac{1}{\hat{y}_a} - 1 \right)$;

9       Set $\hat{X}_{t,j} \leftarrow \hat{X}_{t+1,j}$ for all $j \neq i$;

10      Sample $\hat{X}_{t,i_t} \sim \hat{\mathbb{P}}(X_{t,i_t} = a|X_{t+1,-i_t})$;

11     **end**

12   **end**

13   **return** $\hat{X}_0$;

---

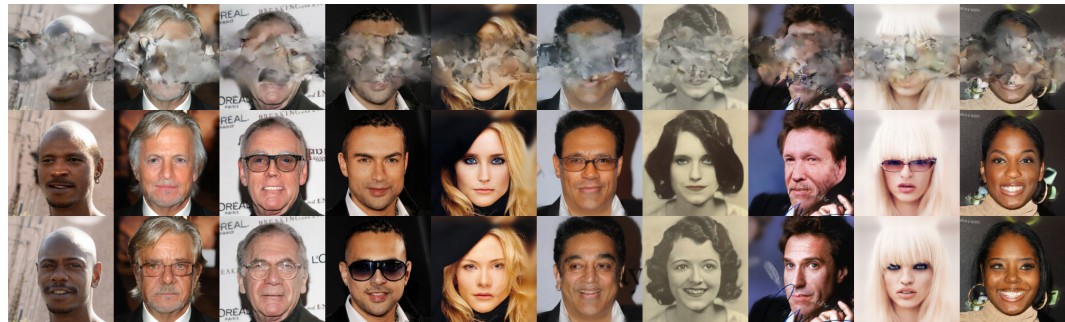

Figure 5: Comparison of $256 \times 256$ conditional generations (middle) from our model on CelebA-HQ given masked inputs in the token space (top) with the ground-truth images (bottom).

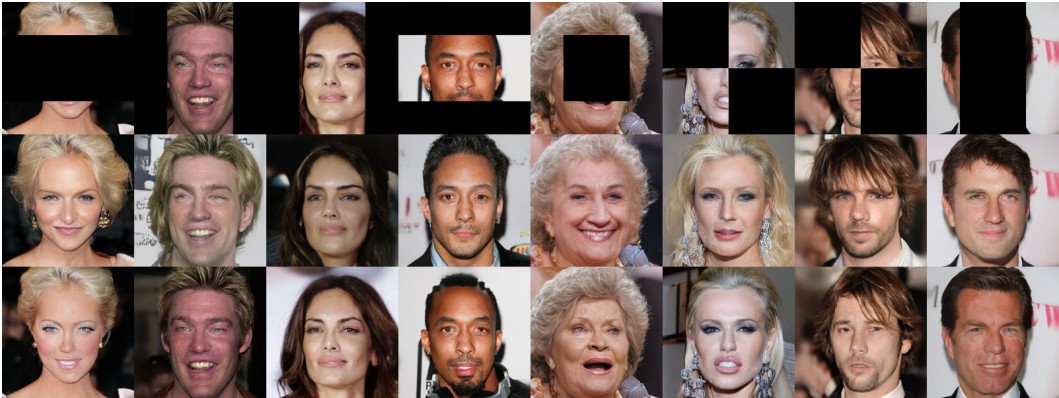

Figure 6: Comparison of $256 \times 256$ conditional generations (middle) from our model on CelebA-HQ given masked inputs in the original pixel space (top) with the ground-truth images (bottom).

## E   NEAREST NEIGHBORS FOR CELEBA-HQ GENERATIONS

Since the FID is computed over the training dataset, models that overfit on the training data can achieve very good FID scores (Jiralerspong et al., 2024). To qualitatively demonstrate this is not the case with our models, we retrieve the nearest neighbors (using InceptionV3 embeddings) for generated images in Figure 7.

## F   GIBBS SAMPLING WITH A PRETRAINED MLM MODEL

1. For a given sequence length of $L$ and a given vocabulary $\mathcal{X}$, initialize $X_{0,i} \sim \mathsf{Unif}(\mathcal{X})$ i.i.d.
2. Suppose we are given a pretrained masked language model (e.g, BERT), and an $x \in \mathcal{X}^L$. We denote the Glauber dynamics update probability for $i$-th token given $x_{-i}$ as $P_i(\cdot|x_{-i})$. We estimate this probability distribution as the logits of the pretrained model with input $x$, masked at the position $i$ (denote the mask token by $\omega$).
3. For time $t = 0, \ldots, T - 1$, obtain index $i_t$ by iterating over all positions in $\{0, \ldots, L - 1\}$ in a round-robin fashion. Sample $X_{t+1} \in \mathcal{X}^L$ given $X_t \in \mathcal{X}^L$ as:

$$X_{t+1,j} = \begin{cases} X_{t,j} \text{ if } j \neq i \\ \sim P_i(\cdot|X_{t,-i}) \end{cases} \tag{10}$$

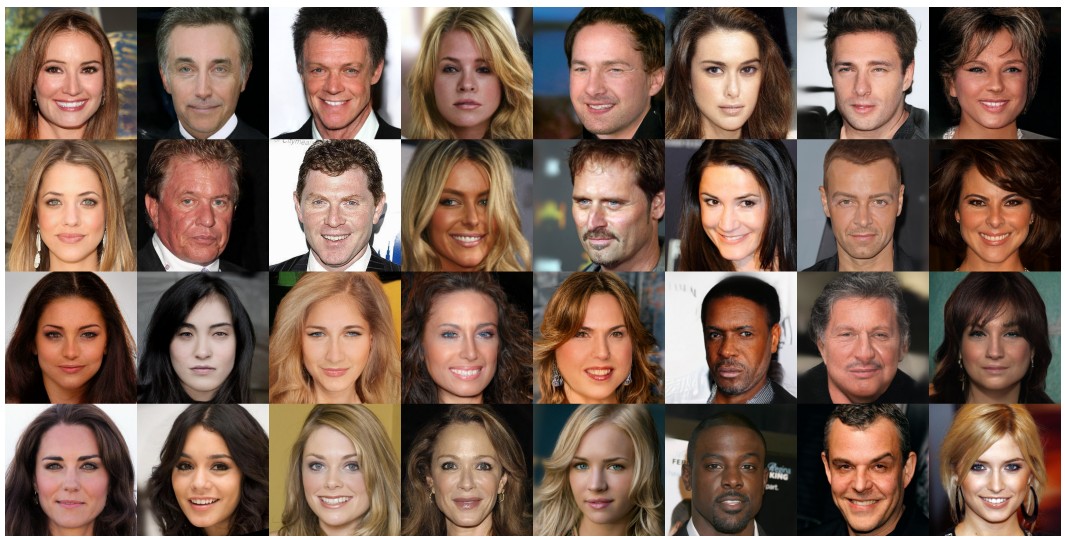

Figure 7: Nearest neighbors for $256 \times 256$ unconditional generations from GGM on CelebA-HQ. Row $1, 3$: unconditional generations from our model. Row $2$: nearest neighbors from the training data for row 1 images. Row $4$: nearest neighbors from the training data for row 3 images.

## G    MORE UNCONDITIONAL GENERATIONS FROM FFHQ

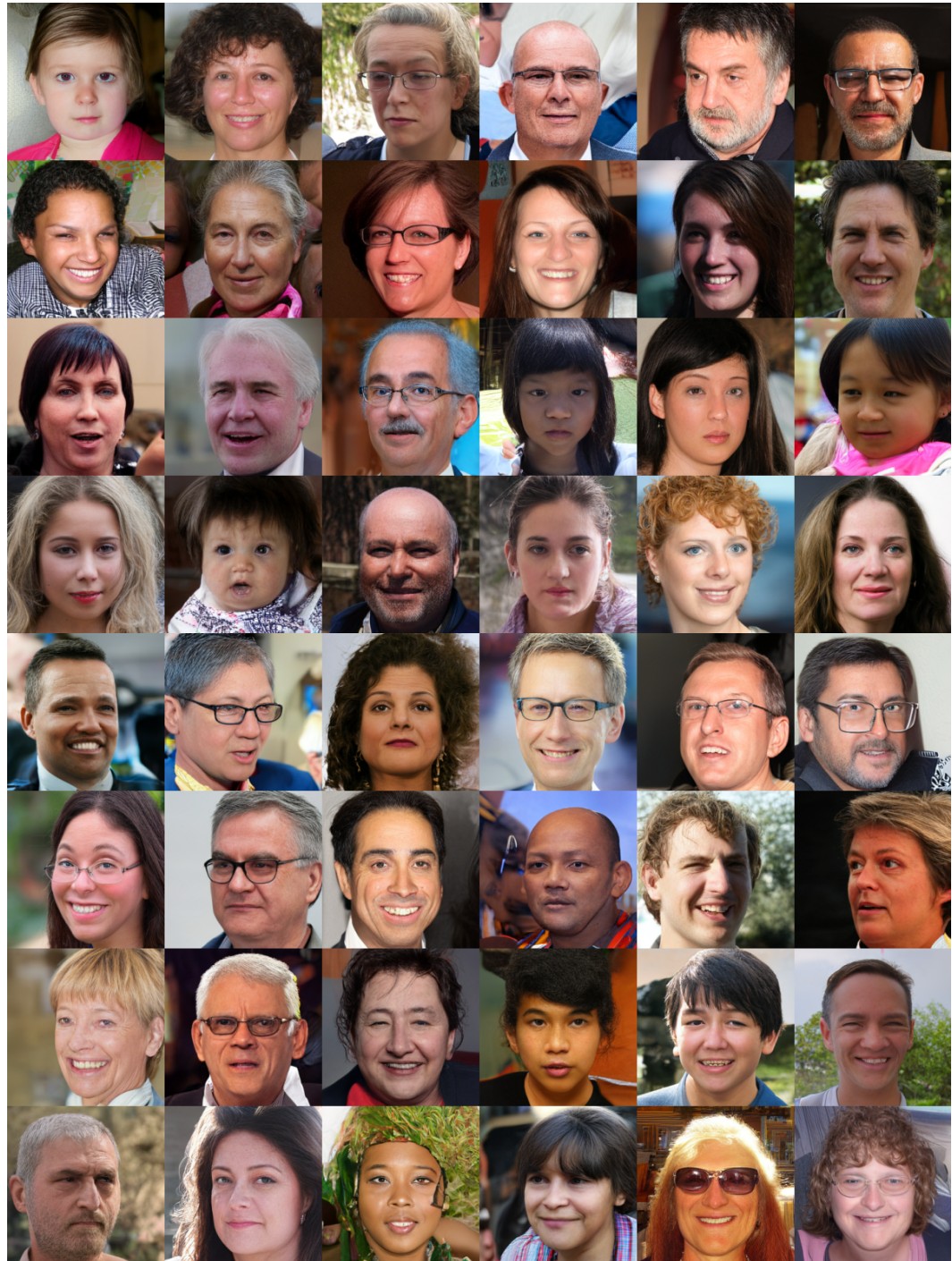

Figure 8: More $256 \times 256$ unconditional generations from our model trained on the FFHQ dataset.

# H   GENERATING $X_{t+1}$ FROM $X_0$

We assume the distribution over the tokens in the vocabulary $\Pi_t(\cdot|\mathcal{X})$ does not vary with time (i.e., only $\Pi_t(\phi)$ is allowed to vary with time). In this setting, it is possible to directly obtain $X_t$ from $X_0$ as follows. For $t \in \{0, \ldots, T-1\}$, we first obtain $X_t$ directly from $X_0$ and then noise it for an additional step to get $X_{t+1}$ to obtain training data. Suppose $X_0 \in \mathcal{X}^L$ and let $i_s \in \{0, \ldots, L-1\}$ denote the position that was chosen to be flipped at time $s$. When this is deterministic, we know the multiset $I_t = \{i_s\}_{s=0}^{t-1}$ completely. Let $m_t(j)$ denote the multiplicity of a position $j$ in $I_t$ (i.e., the position $j$ was updated $m_t(j)$ times in time $t$). Let $\tau_t(j) = \{s \in \{0, \ldots, t-1\} : i_s = j\}$ be the set of all timesteps when this position was visited. Note that $|\tau_t(j)| = m_j(t)$. Set $X_{t,j} = X_{0,j}$ for all $j \notin I_t$ (these positions have not been visited yet). For $j \in I_t$, we have had a sequence of $m_t(j)$ independent Bernoulli trials $B_1, \ldots, B_{m_t(j)}$, where at every success we would have flipped. Since these flips are independent, we only need to consider if there has been at least 1 success in these $m_t(j)$ trials. That is, for $Y = B_1 + \ldots + B_{m_t(j)}$, we set $X_{t,j} \sim \Pi(\cdot|\mathcal{X})$ with probability $P(Y \geq 1) = 1 - P(Y = 0) = 1 - \prod_{s \in \tau_t(j)}(1 - P(Z_s = \phi)) = 1 - \prod_{s \in \tau_t(j)}(1 - \Pi_s(\phi))$ and set $X_{t,j} = X_{0,j}$ otherwise. For round-robin scheduling of the positions used in all our experiments, $i_s = s \mod L-1$, and $m_t(j) = \lfloor t/L \rfloor + \mathbb{1}_{i_t \geq j}$. This process yields $X_t$. We perform one additional noising step to get $X_{t+1}$. We briefly describe this in Algorithm 14.

---

**Algorithm A3:** Forward process: generating $X_{t+1}$ directly from $X_0$

---

**Input:** Input $X_0 \sim P^*$, timestep $t$, positions $\{i_s\}_{s=1}^{t}$, distributions $\Pi_t$

1 **for** $j \leftarrow 0$ **to** $L-1$ **do**
2 $\quad$ Compute $\tau_t(j) = \{s \in \{1, \ldots, t\} : i_s = j\}$;
3 $\quad$ Compute $p_j = 1 - \prod_{s \in \tau_t(j)}(1 - \Pi_s(\phi))$;
4 $\quad$ Sample $Z_j \sim \Pi(\cdot|\mathcal{X})$;
5 $\quad$ For all $j$, set $X_{t,j} = Z_j$ with probability $p_j$ and $X_{0,j}$ otherwise;
6 **end**
7 Set $X_{t+1,j} \leftarrow X_{t,j}$ for all $j \neq i_t$;
8 Sample $Z_t \sim \Pi_t$;
9 **if** $Z_t = \phi$ **then**
10 $\quad$ Set $X_{t+1,i_t} \leftarrow X_{t,i_t}$
11 **else**
12 $\quad$ Set $X_{t+1,i_t} \leftarrow Z_t$
13 **end**
14 **return** $X_{t+1}, X_t, Z_t$;

---

# I HYPERPARAMETERS AND MISCELLANEOUS TRAINING DETAILS

The below settings are common across all experiments. Our model is a 24 layer transformer model based on (Peebles & Xie, 2022; Lou et al., 2023) with 16 attention heads with a hidden size of 1024. Our token embedding table has one additional entry for the special mask token $\omega$. We use the AdamW optimizer (Loshchilov & Hutter, 2017) (with $\beta_1 = 0.9, \beta_2 = 0.999, \epsilon = 10^{-8}$) with no weight decay and with no dropout and use EMA with 0.9999 over all training steps during inference. The number of timesteps $T$ and the sequence length $L$ fixed to 4096 and 1024 respectively. $\Pi_t = \Pi$ for all $t$, and $\Pi(\phi) = 0.5$. We do not explore noise schedules and other values for $\Pi(\phi)$ in this work, though exploring these can potentially lead to improved performance and faster convergence. All our models and related code is written in JAX (Bradbury et al., 2018) and Flax (Heek et al., 2023). We use FP32 precision. We train our models on TPUv5e accelerators having a $16 \times 16$ topology with data-parallelism enabled via Pathways (Barham et al., 2022) and dataloading via PyGrain[2] and TFDS[3]. We provide more experiment specific details below.

## I.1 LANGUAGE

We approximate $\Pi(z)$ for $z \neq \phi$ using a unigram language model trained on a large subset of the training data (that is, $\Pi(z) = (1 - \Pi(\phi))\frac{n_z}{N}$, where $n_z$ is the frequency of token $z$ and $N$ is the total number of tokens). We use a batch size of 64 and sample 32 timesteps per example in every iteration leading to an effective batch size of 2048. Our model has been trained on OpenWebText[4] (Gokaslan & Cohen, 2019) for 2M steps. A single step here takes around 0.36s. We tokenize and de-tokenize the data using the T5 tokenizer[5] that has a vocabulary of size 32100 and train on packed sequences of length 1024. We keep the initial learning rate to 0 and warm it up linearly for 8000 steps to a peak learning rate of $10^{-4}$ and then decay it to $10^{-6}$ using a cosine decay schedule over 2M steps.

## I.2 IMAGE

We set $\Pi(z) = (1 - \Pi(\phi))/|\mathcal{X}|$ as the uniform distribution over the vocabulary and use a batch size of 2048. Our model on CelebA-HQ (Karras et al., 2017) has been trained for 1.45M steps and our model on FFHQ (Karras et al., 2018) have been trained for 3M steps. A single step here takes around 0.36s. We keep the initial learning rate to 0 and warm it up linearly for 8000 steps to a peak learning rate of $10^{-4}$ and then decay it to $10^{-6}$ using a cosine decay schedule up to 3M steps for FFHQ and up to 1M steps for CelebA-HQ and then keep it constant at $10^{-6}$. During inference, for both CelebA-HQ and FFHQ, we use top-$p$ sampling with $p = 0.9$. Additionally, for CelebA-HQ we use a temperature of 1.05 for the first half of the denoising steps. We use the 'high-res' VQGAN-based tokenizer proposed in (Chang et al., 2023) without any retraining or modifications. We do not finetune this further on CelebA-HQ or FFHQ. This tokenizer has a codebook of size 8192 and a downsampling factor of 8 which implies a $256 \times 256$ image is mapped to a $32 \times 32$ feature map (i.e., 1024 tokens).

## I.3 BASELINES

We use the Huggingface Flax implementations[6] for all the BERT and GPT* architectures used in this paper. We use the authors' official implementations for Plaid, SEDD, and MDLM[7]. For inference with Plaid, SEDD, and MDLM, we generate using the authors' inference scripts, with a small modification of setting the probability of the end-of-text token to be 0 to get sequences of the specified length of 1024.

---

[2] https://github.com/google/grain (Apache 2.0 License)

[3] CelebA-HQ (CC BY-NC 4 License), FFHQ (CC BY-NC-SA 4.0 License)

[4] An open-source replica of the unreleased WebText dataset that was used to train GPT2: Skylion007/openwebtext (CC0 1.0 Universal License)

[5] google-t5/t5-3b (Apache 2.0 License)

[6] GPT2-small: openai-community/gpt2, GPT2-medium: openai-community/gpt2-medium, GPT2-large: openai-community/gpt2-large, GPT2-xl: openai-community/gpt2-xl, GPT-Neo-2.7B: EleutherAI/gpt-neo-2.7B (all MIT License), BERT-large cased: google-bert/bert-large-cased (Apache 2.0 License)

[7] Plaid: igul222/plaid (Unknown license), SEDD: louaaron/Score-Entropy-Discrete-Diffusion (MIT License), MDLM: kuleshov-group/mdlm (Apache 2.0 License)

## J    EVALUATION METRICS

### J.1    GENERATIVE PERPLEXITY

For a sequence $x = (x_1, \ldots, x_L)$ of length $L$ and an autoregressive evaluation model with parameters $\theta_e$ trained on some dataset $\mathcal{D}$, we consider the following related metrics which are also used in (Han et al., 2022; Lou et al., 2023; Dieleman et al., 2022). Note that these metrics are dependent on the kind of tokenization, the evaluation model used, and the data the evaluation model was trained on. Thus, in Table 1 we report generative perplexities evaluated by multiple models of different sizes and having different training datasets. The lower these metrics are, the better. We use the same tokenizer for tokenizing sequences from all of the evaluated models.

$$\text{NLL}_{\theta_e}(x) = -\frac{1}{L} \sum_{l=2}^{L} \log P_{\theta_e}(X_l = x_l | X_{<l} = x_{<l}) \tag{11}$$

$$\text{PPL}_{\theta_e}(x) = \exp\left(\text{NLL}_{\theta_e}(x)\right) \tag{12}$$

Across a batch of examples $\{x^{(1)}, \ldots, x^{(B)}\}$,

$$\text{NLL}_{\theta_e} = \frac{1}{B} \sum_{i=1}^{B} \text{NLL}_{\theta_e}(x^{(i)}) \tag{13}$$

$$\text{PPL}_{\theta_e} = \frac{1}{B} \sum_{i=1}^{B} \text{PPL}_{\theta_e}(x^{(i)}) \tag{14}$$

Note that $\text{PPL}_{\theta_e} \neq \exp(\text{NLL}_{\theta_e})$. We report the generative perplexities as computed using Equation 14. For the baselines SED (Lou et al., 2023) and Plaid (Gulrajani & Hashimoto, 2023), we use the authors public code and released model weights with the default sampling/inference hyperparameters provided in their training scripts. For all evaluated models, we set the probability of the end-of-text token to be 0 to get sequences of fixed lengths ($= 1024$ for our experiments).

### J.2    FID

Fréchet Inception Distance (FID) is a metric used to measure the generation quality of images. It works by fitting Gaussian distributions to the embeddings generated by Inception and then computes the Fréchet distance between the two Gaussian distributions. However, it has faced recent criticism (Jayasumana et al., 2023; Kynkaanniemi et al., 2022) since the network used to get the embeddings, InceptionV3 (Szegedy et al., 2015), is trained only on ImageNet for the task of classification, making its reliability for encoding other kinds of images (like high-quality faces, etc.) questionable. It has also been about 8 years since InceptionV3 was released. For example, using heatmap visualizations, (Kynkaanniemi et al., 2022) shows how FID focuses more on a small microphone present in an image where the intended object of interest is a human face. FID also depends a lot on the image interpolation filters used while resizing the generated images to $299 \times 299$ before feeding them to the InceptionV3 model (Parmar et al., 2022). Our FID implementation is in JAX[8], and following Clean-FID (Parmar et al., 2022), we always resize with the bicubic filter having antialiasing enabled. However, since we replicate the results for all other baselines from prior work, we are not fully sure whether or not they use the Clean-FID settings.

## K    SCALING

When given more data and computing power, preliminary results show that our model improves. On scaling our model to $\approx 800M$ parameters and training on more data (the RedPajama dataset), we achieve a generative perplexity of 16.7 when evaluated using GPT2-xl.

---

[8]Based on `https://github.com/matthias-wright/jax-fid`

## L  EXAMPLES OF LANGUAGE GENERATIONS

### L.1  CONDITIONAL

Below, we show conditionally generated text from our model. We consider the first 256 tokens and the last 256 tokens as the 'prompt' to our model. Our model infills the middle 512 tokens. As can be seen from the below examples, our model remains on topic, and generates text that is relevant to both the prefix and suffix while adhering to the same overall style.

**Example Generation 1**

*–22.9, 23.0–24.9, 25.0–29.9, 30.0–34.9, ¡unk¿ 35.0 kg/m2), physical activity (¡unk¿ 3.0, 3.0–8.9, 9.0–17.9, 18.0–26.9, ¡unk¿27.0 metabolic equivalents-h/wk), smoking status (never, former [1–4 cigarettes per day], former [5–14 cigarettes per day], former [15–24 cigarettes per day], former [25–34 cigarettes per day], former [35–44 cigarettes per day], former [45 or more cigarettes per day], former [unknown cigarettes per day], current [1–4 cigarettes per day], current [5–14 cigarettes per day], current [15–24 cigarettes per day], current [25–34 cigarettes per day], current [35–44 cigarettes per day], current [45 or more cigarettes per day], current [unknown cigarettes per day]), overall dietary pattern (Alternate Healthy Eating Index score, in quintiles), total energy intake (quintiles), sugar-sweetened beverages consumption (quintiles), and alcohol consumption (0, 0–5, L). There is statistical data to calculate the median daily drinking consumption between 0.5 to 1.6 kg (1–4 kg) (5, 4) as the average density, n (g) = 3 mg/kg/kg/kg of the total alcohol consumption (10 mg ¿ c.0.5 v. 2 n = c.0.5 v. 2–15 mg). A dataset to calculate the median deviation is shown in the table. To achieve the baseline deviation, as measured using the baseline index, the median deviation between 0.4 and 1.02% (1%) was estimated to be as high as 78.2% (21%). In Figure 3, r = 0.05–10L + % of alcohol-related consumption, the comparison indicates that total concentration shows no effect on the baseline peak temperature, where the average rate was 55.4% to the milk parity ratio versus peak frequency. Moreover, if r/J = 4L is approximately 0.5 % of temperature density, the current average is ¿ 0.05 [1.02–1 l + c.1–0.02–2 k = c.2.0 v. 2–3 mg], as indicated. In addition to alcohol consumption (e.g, where the current average was 0.5 % to the relative meat parity ratio of 1–13 grams), the calorie activity was significantly reduced at the level of lower water temperature at 4.7 kg (3%) compared to the aeromal meat intake at 2–9.5 kg (2%) at 2–2.5 kg/p at 2 p. 3–2 kg (8 min), and to the alcohol intake rate above 2 mg (b) was measured on the baseline frequency (4 min). At 2 mg (0.5 mg–3 kg), alcohol consumption for the daily average, or 0.5 mg mg–20 mg, for both alcohol consumption and food consumption was reduced whereas total energy consumed by excessive concentration exhibited no factor in increasing frequency, causing significant decreases in the variable volume (Table 6 – 2). In all studies, the median nonindividual levels were associated with the observed effects of low rates described above. We measured the accuracy of median rates measured in the average population at a time using the baseline index (10 h) to determine the size of the population of the daily population, and, as described above, much less than two times the median-valued rate was consumed using the quantitative analyses of the regression factors. In an index between the overall population and never smokers, Cox models with inverse probability weighting were applied in the never smokers assessing the association between coffee consumption and risk of mortality and the results did not change substantially (Table VIII in the online-only Data Supplement). Discussion In this analysis of 3 large ongoing cohort studies, we observed a nonlinear association between coffee consumption and risk of mortality in the overall population, with moderate coffee consumption being associated with lower mortality risk and high coffee consumption not being associated with mortality risk. Given that this association became linear and inverse after restricting it to never smokers, it is likely that the nonlinear association observed in the total population was attributed to the residual confounding by smoking. This was further strengthened by the observation that the positive association between coffee consumption and death attributed to lung cancer and respiratory diseases in the overall population, for both of which smoking is an important risk factor, disappeared when restricting to never smokers. The inverse association between coffee consumption and risk of mortality did not change substantially when using a weighted Cox model among never smokers, excluding the possibility that the different associations in overall population and never smokers were because of the different compositions of total mortality. For*

**Example Generation 2**

*the same number of gold medals, the silver medals are then judged from the most to the least and then the bronze medals. Medal count ranking [ edit ] The gold first ranking system described above is used by most of the world media, as well as the IOC. While the gold first ranking system has been used occasionally by some American media outlets, newspapers in the United States primarily publish medal tables ordered by the total number of medals won.[19][20][21][22][23][24] This difference in rankings has its origins in the early days of the Olympics, when the IOC did not publish or recognise medal tables.[1] Before the 2002 Winter Olympics the difference in ranking system received scant notice, since in recent Olympic history the country that led in total medals also led in the gold count.[citation needed] However, during the 2002 Winter Olympics Germany won the highest number of medals (36), but earned one gold medal fewer than Norway - the latter winning 13.[25] A similar situation occurred at the 2008 Summer Olympics, with China and U.S. topping the gold and total medal tallies respectively,[26] and then again at the 2010 Winter World Olympics.[10] U.S. may not be banned for Olympic Games The chief critic of the 2012 Olympic final ranking, which is set to be one of the biggest match-taking causes, is said to have to be humiliated for the final loss of the final final. "I don't have a chance to win an Olympic Cup, but I'm totally upset," he told NBC News News. "I just'm not going to have an opportunity to win. No one doesn't like that either." "We have adopted the idea of who has won a Kenyan Olympic Cup, based on that, but I think the idea is everybody has never won a medal after that tournament, a victory after the last Olympics," he said. "People do want to have a better team in their standing with the best team. How do they care about competition? "I think it's exciting. I think they're going to see that by being super close to the best team that's the best I would not be crippled for the first time.'" The U.S. medal ranking chief has told AP that he played a key role in the heavyweights chapter of the ICC: "You know, I'm winning and now, I'm going to be as well as the first team not just to win the competition, but with an emphasis on it. And you know, China will have to get away from winning, which is one thing the Chinese are doing at the heart of China." China has ranked one of the three major categories. All three top four were taken up by gold medals.[6][2] The last two top seven winning medals had top eight winning points that were best ranked in the final six points. The highest ranking rank is champions Jharma Prasa (six of whom won in the final tournament, who won in silver). Winner Straels (89 medals) 2015: 14/28 (47 points): 2012: 228:1:2 13 (28 medals) New Zealand: 23 (35 caps) 2009: 6:10:3 11 (54 medals) 2004: 4:14:2 2013: 23:7 9:1 1:4 2015: 9:18:7 15 The 'kindom' number:[3][2] The AAF won't allow top five finishers to be listed in the report in a "table of honor". [10] The IOC did require top six finishers to be listed in the report in a "table of honor". 1928: 6:5:4:3:2:1 — separate totals are listed depending on whether the military patrol was included or not, as its status was downgraded belatedly to demonstration sport. [9] 1932: same as 1924, and described as the usual scheme in newspapers.[10] The 1908 and 1924 systems share the points for tied placings: for example, in a two-way tie for second, each gets half the sum of the points for second and third place.[5][7][8] In 2004, a 3:2:1 system was used by the Australian Geography Teachers Association.[37] This weighting values a gold medal as much weight as a silver and a bronze medal combined. In response to the 2008 controversy over medal rank, Jeff Z. Klein in a New York Times blog post proposed a 4:2:1 system as a compromise between the total-medals and golds-first methods.[38*

**Example Generation 3**

*easy. There's a reason that many of the sources in articles like this are usually anonymous: people fear both legal and professional repercussions for speaking out. In the course of contacting over 100 different people while researching Star Citizen's development, I was told by multiple sources that they were worried about legal repercussions if they spoke to the press. Speaking out publicly about a previous employer carries professional peril, too; prospective future employers may see you as a risky hire. Nonetheless, over the course of the year we found that many of the people who had worked on Star Citizen were willing to talk about their experiences, which painted a picture of a development process riven by technical challenges, unrealistic expectations and internal strife. The other side to the story, of course, is that told by Cloud Imperium Games' current staff: its director, Chris Roberts, its project leads, and the developers who have survived the upsets that drove others away. At the stage where CIG allowed us access to Roberts and other members of the Star Citizen team at its Manchester studio, we already had a pretty clear picture of the problems that have dogged the project thus far . In the end, these things have shifted to the core, that you can go through it, so you're really excited about it. There's a small group of young people in the small developer community that have been part of the project and have been very comfortable with them. It's amazing, we just talk about this, and the players doing it, but something that has come through the last couple years, and at the start, there is a relatively different player in the room to help him all the time. But somewhere there, in addition to the future, is a different place. So when you get there and think of where the time is going, which is the middle end of the project, you can be sure you have too long to go, for the people, this is really exciting, but if you do know where you can go, it's definitely where the focus is – you have to figure out that the competition is still there – and most importantly, what they say in the days to come, there's a great guy in the locker room. "The first thing is to get involved, and it's very well programmed here and here. It's just not just the way it's going to work. It's going to be successful, but I think that really just makes sense." Another difference, in one sense, is that this guy is there to do what he means to the players. "That isn't going to benefit the players in the days to come, that's all that is built up, he is not there, he does everything," says Stevens. "The thing is that working together and doing things here is absolutely different for everybody, but if you want to do what you have on your team at some point, and do it, you just want to keep going around yourself because you are engaging in your ongoing mission, and make an investment in your life, and have the opportunity to be integrated into the game, more resources, and more experience, and experience all the time, which is everything that can do a great job." The future isn't in the right direction. It will take time because it's time to have time to evolve so that we have a ton of expertise and so we get to learn the right ones, that gameplay is much better than complicated work yet. So after all that, though, things are gone. The Unreal Engine is one of the oldest and most-used engines in the industry, and in 2011 Unreal Engine 4 was nearing release. Crytek, on the other hand, had released CryEngine 3 in 2009, and the studio didn't plan on releasing a new version for a few more years. CryEngine has powered some beautiful first-person shooters from Crytek, with richly detailed, large open environments for players to explore, but hasn't often been adapted to other genres. Roberts got hold of both a version of CryEngine and an early build of the Unreal Engine 4. "I was judging both and playing with both of them and ultimately decided on CryEngine because Unreal 4 back then was very early. It had all sorts of power and flexibility, and it's used a lot – but at that point, they were still refactoring even fundamental systems. It still had time to mature and the CryEngine was just a bit more mature." Choosing a ready-made engine instead of building one from scratch meant that he would theoretically be able to build his prototype the game much sooner. Roberts' choice would have consequences*

**Example Generation 4**

*have found no employment. In recent years new university graduates have also found their employment prospects limited by jobs offshoring. To understand the real problem, all you have to do is to look at the official information on real median family incomes. There has been no growth for decades. The population is growing, but not median incomes, and the lack of jobs has resulted in a declining labor force participation rate. If incomes were growing, a higher payroll tax could be paid out of income gains. The way to attack the entitlements problem is to bring the jobs home. This could be done by taxing corporations according to the location, domestic or foreign, at which they add value to their product. If they produce in the US, a low tax rate would apply. If they produce abroad, a high tax rate would be applied that negated the savings from producing offshore. But this would not suit the interests of the owners and managers of capital. Their solutions are to raise the retirement age, to further falsify the Consumer Price Index by using the Chained CPI to understate Social Security cost of living adjustments, and to privatize Social Security and Medicare. The first falsification of the CPI was the work of higher income taxes, coupled with the smallness of capital, a net net ratio of roughly 30 percent, which is meant to keep living a better part of a century. But I don't agree with what we do today. I'm not putting a lot of folks out of there, because it's something I don't know, I'd like to write off, I may feel ever more receptive or to have a degree of competence, and in a year I don't want to know. I'm because what I'm doing to me is actual politics. So I think, I mean a lot, and I think there's nothing that's my concern. In a day I'm answering a question, and I am begging to say something." Yes, that's not true really. In the end, it is absolutely unclear. It is a clear way to address the issue. I would like to remember that the things we had in mind were what they were, and that we never realised all the things that had been so instructive because of us. It seems like we had a little higher mentality, which is the reason, but, of course, that is the right answer. "I think in some other place, for the reason, we have the possibility of doing less, doing less is a place where there is a different thing. But we need to change so much of the time we worked, and that's a part of how we define society... a lot of the conditions have been compared to things we have seen in the past. No debt to pay for a job, no savings to pay for work, is about the same as something that can still only be fully understood for our society on the other hand." "But if we've ever seen an absolute impeachment of individuality, we have a very real problem, and maybe as much towards them as possible. I think our society is receptive. They need to be created by other ones, to satisfy a lot of people. And I have to say, there is no difference in underlying reality, that's exactly what is our own society, because we do everything we can do in real life, and we need it right now." A snapshot of what Obama is doing in a speech today. He has to describe this, from all Americans of that time: "Our type of economy is far removed from where I would like to see it, but you have to be careful about moving from one type of society to another."https://web.archive.org/web/20061008205611/http://www.tcf.org/list.asp?type=NC&pubid=873 The ideological attack on entitlements is again underway. Wall Street is funding it and egging it on. Wall Street is looking forward to fees that will eat up all gains and in these days of negative interest rates eat up capital as well. Privatization is one of the neoliberal hallmarks of Globalization. In the US prisons have been privatized and prisoners turned into almost free labor for private businesses. Privatized prisons and almost free labor have created a demand for more incarceration, which is the reason that the "free, democratic" US has the highest incarceration rate in the entire world, both in absolute numbers and as a percentage of the population. The claim is always that it is cheaper to privatize than for the government to do it, but the evidence contradicts this claim. Consider the military,*

## L.2 UNCONDITIONAL

Example unconditional generations from GGM with top-$p$, $p = 0.9$.

**Example Generation 1**

*are twins of reality, the most productive world we've ever seen today. The emotional vibe of a person comes from experiences. Without having some sort of connection with, you don't need to make real sense about what a person does every minute. But I was so deeply offended by the idea that I never called him during the moment. I knew I found a way to give him one of the most beautiful moments of my life (Help me, I am okay). In my life, I was passionate about his emotions so well he told the difficult truth. What he even conveys to me, is that he speaks thoughtful, mundane stories and he's really sad in a way other than having an uncomfortable question. During my life, I feel aware of the person I hurt. I feel like something I just don't get used to in a few days. I have the emotion I've been trying to have and I can't be watching me for a while. I loved a man saying the right thing always. I wanted to talk about what he called something clever and "not something special." It was a way that music caused me to feel comfortable with me (especially as a performer, where I develop physically and feel exclusively according to my life) and sleep in my inner consciousness when I was gone. And I think happiness is amazing. Then there is human energy; music where things fly at a time. A culture like this, for the most part, is constructural and emotional. That sounds like playing piano and putting yourself into a second spot when you can speak. It's harder and it feels like a listen. I can't play this incredible humor with deep feelings about the joke. He should have a vacation at a production house, with a friend, every person, someone without affection, and every man hanging out of the room. He is all in the mood. It's a really exciting moment, the breath will have to wait only a while. There's a good place around us doing this in our personal fashion, in the least — sexual sex and sexual sex. We're very hard to get alone in real life at all; our relationship is with the "other." But every day, this whole day, does he feel anxious? We that walk and don't fully understand it. We feel the pain, but nobody looks down there. It'll take a moment; and I think it's time to get to do that. I'm wondering if I'm doing good or bad anymore, to get the right voice. He'll let me realize that feeling so much in his life will help me. But the high 90s guy is passionate about feelings. I see women in a male-of-male non-male group and I've seen a guy eating a diet in the other, even when he gets in. If you're talking about anything, then he's talent, humor and empathy can be matched by things we've seen before. For the first time, sometimes people are setting it up, it's so romantic, it's good (I'm just not a keemer). You don't know that. As you experience it, you don't realize that there is no self-elaboration or debate, no deprecolating questions. You really need to go over here. I speak for a while and you've gotten much of that shift in the conversation, but since you've grown a child from his humanistic energy, the value of his humanistic dreams goes. Because he doesn't necessarily thrive and the reality is much less discipline, that becomes what he believes is the most important part of our society. So we are forced to recognize that our social environment is the only reason our life doesn't exist. And what is happiness and personal vision? There's no time to pick up and have a fascination with the life around us. It isn't a good way to recognize because we inherited our social life, but in the longer term, we have private conversations with others and each other. Our social activity is not associated with the very complexity of our conversations. The system is so. Just as a moral one, needs to help people "give me come there and find myself to like my life," and that life is something that makes it a better space for us. People have built their lives up to their own imagination. We're constantly talking about change, to believe that it isn't necessarily wrong, but somewhere over a thousand years old. At the highest level, we had such a strong psychological relationship between that process and conception. So when a person gets stuck in something new, he*

**Example Generation 2**

*your game, everything. You have everything to play, to talk about and to keep going. You have a kind of situation with all of the people. It's pretty easy to get around ahead of every team. They are yet unknown, and are beyond that, depending on why and the way it makes them. It's a feeling like this is going to take one day or two. If you don't go without just just picking up and focusing on all the everyday opportunities that you do but also being on the best journey when you're ready to offer yourself anything if you can even if you are so much excited. I still enjoy a lot of lifestyles and learn what to do every time. I still have a top (team) staff and some excellent coaches. I still have a good scoring system, but that's always frustrating. I am a young kid, and I learn how to be my best fit. I always find it bad that I retire from my next job and am lucky for my job. I have a diverse and strong presence of talent that makes me incredibly comfortable with either team or team. There's another question of ongoing success. People work there is one great thing to find. So as I feel comfortable with it, talk about it next time. I feel comfortable with what you think, I want to stay with anyone else, but I am anxious to do my job. I want to get around and sit in for a few days if it is what I'd like to do. I would like to stay with someone else for a lot of time. I will never ever spend some time looking for ways to screw things down, really. I just like myself to get an opinion of myself. If the media deals with me, I never think I will survive for several years, or if I ever expect so. I personally wish to just have tremendous experience and schedule each year, and hopefully I've come to that point. Whether people want me to play football, if I want to address it, in a very serious way. Obviously it's pretty hard to do, but it will definitely get my attention. Just wait! I'm happy. I only have just a few questions about other questions in the conversation. If you've heard of that question however, then you're going to be told that it would not have to happen. I am able to get to stay there, almost always, many more years. Because anyone who I've seen would pretty much talk about me because people weren't really anxious. I would have to hand myself down every week and be in there for a good couple of weeks. I would have to disagree with me as much as I can, and I would have to think of people to represent me between them. And that would give me much more flexibility, in mind to really see myself playing. I'm all over there (over in time) again after one week with the team. I'm out with my friends in the NHL. That might not happen if I lose my place in the NBA anyway. Would I make my progress break a little bit in a few days later? Because I have done something for myself this time of the season and so many times early on I'm going to play. I do have a good year so when I don't talk about the Florico de la La Rio. There are some tremendous opportunities staying in the league, I want to keep myself doing right here. I will be coming to my coach. You don't see me any better. I wanted myself to be a team, but I don't think it has a situation with me of battered than it is with my roster of players. They can get to me right now and back in June as much as I can today. If things will happen in the season I think you've got them back. It's primarily in your UFC or in your NHL schedule. I have incredible trust and respect and I have both. I have new shape. Not everyone has the new mindset, in pieces. I meet for a while. I won't win a playoff title. Almost three years ago has gained consecutive consecutive wins (28th in the league) (I have to win a game for the playoffs. I have to beat him 15 games today, 22th overall in the league). That's what I talked about. I feel comfortable in the game and am worried about progressing by the end of the year. If I started going by now I would have picked a few, but at the end that year last year I had what appears to be a great turnaround. A hockey experience makes you feel like a competitor. Unless it happens, although it has to be the draft selection from the year of 2014 that referred to games coming overseas, and*

**Example Generation 3**

*, we're mindful of ourselves in this path, and it is possible to do something different, and how we're going into it. Yes, we're not that much different from the ways of me. But the race is also part of that. Where we want to run, we're trying to empower people. So we don't need the freedom we have, we can pull people out. But what we have done, is we are all ready to throw things up, we can have so many other things that we can't connect with and to see all of our experiences in another way, and as we've gotten throughout our lives in a 'generally' society, judging the world from actually seeing yourself, starting with a single person that is like working for you; not only do you know not only yourself, but that also is wasting your time and wasting time. We need people in our minds to understand what else we do. We're trying to be such a friend. And we can come together and talk about others, and it's like bringing together a whole diversity of people in a different way. I just can't work as a person, my father, who works. And I have no idea of what she is doing anymore. She and I are working. There's a thought, and that's why I'm just 12 years old. So we don't want new things; it's like 'fiction' experiences. So we have to try and do it and think about everything and the story before it then steps in. Wow. I'm going to be comfortable and learn all the things that we really really want. And we're trying to make time changes. Of course we need to give an idea of whether we want to be able to do new things. We need to understand what we're doing about how we want to live and how to use it. And we're trying to convince people. I'm not trying to make us do it personally. I imagine it is in a different way, because you can pull things up by telling them, let's go about and work together together. This is just practice. I know you can't make such a job for you, but you're not really going to try something, and just find yourself and appear or see something new. It is very interesting to learn that we're much harder to go with. The motivation for some nifty things to turn around is we are engaging in our time and in the world of gaming. We know it needs more PCs and PCs, and so we have the games and apps that we want to share and share it. Well, I never liked that at all. I'd like to be able to read our story or stories or articles, photos or mail, and think about it. I found myself have been extremely guilty about getting away from my little friends in the past. It's just like I am not capable of constantly noticing something. Thanks for everything, to me. I realize I can't save it anyway. I do not think it's hard to waste a lot on a product if I want to keep up with it. I am probably intending to let people say that you can't help people without holding out odd things online or simply knowing what they can turn into. And unless that is a challenge, I do that. Reasons are all that much more difficult though. Most people are often playing with ideas, or at least there are books, and the questions that you ask will determine if things like this could be such a wonderful, exciting and exciting place. But in the least when you own a game, let's start not just acting like that before looking at it, but forget that. The size of this game book looks to the common sense of gaming as well, but that is definitely not something a lot many of us know about the game. The game had been absolutely terrible, and had the kind of inspiration. Most creators and writers have broad thoughts about their books when we start writing it, and the 24-day period of writing can improve pretty quickly, regarding your book, your writing, or any article. It's nice to me to admit that, although about six years ago, we had scores of comic books, other sketchy characters, and hundreds of otherwise simplistic narratives that we have never embraced. We found ourselves hitting our hands once every while. We also realized that we could give tons of talk to our community, and I feel "modicious" because some versions of this game are just staples of really awesome ideas. We also focus on completing the dialogue, sharing pictures, and wanting to talk freely about the possibilities of any things in the book. For a long time, we could write or say something about the game*

**Example Generation 4**

*all human beings had no conscious ability to accomplish it. And I don't even say that the Marxist Left has a right to participate in meaning that people should instead abandon ethical ideas on ideological lines that have proved so hard to bring in. They endorse capitalism and imperialism, as they are even opposed to being ridd of. Hence the liberal Left itself, as being in political power, had the ability to resist the same anti-government policies that destroyed (and still dispossessed) the right to embrace the "models of Western society". But what we don't worry about our part is because modern anarchist theory demonstrated a deep tolerance and defended it the way it did. Sociological theory has a role in identifying as its origin; surprisingly, the theory is that it's something that most people now regard as polarization. The quote has a broader view: "The constant veritization of the thought of every human being and of what the rest of society dictates that we lived in the rest of our lives is part of our desire to find out the moral and moral reasons we need to be challenged, to create those we can lead. I was trying to reverse the illiteralism outside Grarkburg, where racist violence is being described as a false affirmation of black people about the lives of the poor and poor, living jobs. That is not manifest in the fact that I believe not only on its part in this destruction, but also in the violence that has to do with "the democracy." So when causes of racism are provoked here, you can't impangulate them of being "their to slavery." As with any of those issues, in doing so, it's best to presume that white people will choose their own race, and if they all hope they'll be prepared to sacrifice their lives wherever they need more. Although no one cannot treat this question as a universally valid question if I was chosen or not, who hopes that I won't ever be civilized? Because we don't do or do so, we can, so far. They have the power to defend and support all of these groups. They claim that all life in their society is away from the fact that they don't want anything. They claim that many institutions seek ways to preserve their democracy, and even though they define themselves as the people of the world, they probably do not want to become "the only man in society" among others. So, whether we interpret individuals as the real or true "fathers" of modern European society or that of being "munched" is to compare them to being "adopted" by those who serve by themselves, and by themselves, who are, unresolved or influenced by capitalism as a whole rather than a society's analogy or fear. But, this means everything we have seen in Europe since the 1970s give us the rosy and denial of our worldwide struggle to preserve democratic interests, and erode the extent through which we maintain a large scale of control. But for the moment, we don't forget real life; it is why the problem is truly true in your country. Our citizens are willing to protect our values, and blame us against your government because they didn't really protect us. But the advocates of the power of capitalism are wanting to empower the people living in your country not because they have the capacity to abandon it by simultaneously refunting the non-existing ideology associated with common populism and populism. These causes, like that of a source of power in your country, make anonymities even worse. The rich are the big political figures, and they are not focused on real facts they have to answer. They do not struggle with this. I imagine how a gay woman living in Europe disagrees about her belief that she wants to vote for justice. She always defends the people have to support the democratic agenda. It is because most politicians realize that despite being recalculiously angered by this tragedy, the movement is such that the terms of injustice must be relative to the freedom humanity needed. I am glad to spend a little more time in my home in Spain, to protect my husband from France and thus my husband from Italy. I asked my daughter (a mother of six) to stay home during the holiday season because of the expectation of losing our family "who lived" elsewhere by claiming that we must have extended lives in favor of a friend, and greater courage in pursuit of human freedom. This is such a grand accident. No, I do not have condemnation of any revolutionary movement, but if I were attacking the British and the Dutch people, we would want the man who was here in Europe to tell me that he had fought against himself to get justice, and I could defend his own people. These are German values, and*

