# OpenReview forum: "Glauber Generative Model: Discrete Diffusion Models via Binary Classification"
_ICLR.cc/2025/Conference — ICLR 2025 Poster_

### Official Review · Reviewer_cyno · 2024-11-04

**Soundness:** 3
**Presentation:** 3
**Contribution:** 2
**Rating:** 5
**Confidence:** 4

**Summary:**

This paper proposes the Glauber Generative Model (GGM), a new discrete diffusion model framework. Unlike existing methods that often rely on regression or variational approximations, GGM frames the core denoising step as a binary classification problem: for each token in a noisy sequence, the model classifies it as either "signal" (belonging to the original data) or "noise" (introduced during the diffusion process). This simplification allows for efficient training using standard binary classification losses and avoids the complexities of estimating transition probabilities in high-dimensional discrete spaces. GGM utilizes the Glauber dynamics, a Markov chain that flips one token at a time based on conditional probabilities, to iteratively refine noisy input towards a clean sample. The authors apply their model to both text and image generation, demonstrating competitive performance on language modeling benchmarks and promising results on image synthesis using a generic VQGAN tokenizer. Notably, GGM outperforms other discrete diffusion models on language modeling while using fewer parameters. In image generation, it achieves decent FID scores compared to GANs and continuous diffusion models, though a gap to state-of-the-art remains. The authors also showcase the model's ability to perform zero-shot text and image infilling, highlighting the framework's potential for controlled generation.

**Strengths:**

- The reframing of discrete denoising as a binary classification task is a novel contribution. This  simplifies the learning problem compared to prior discrete diffusion models. The use of Glauber dynamics, while not new in itself, is creatively applied here within a diffusion framework and offers a theoretically grounded approach to discrete data generation.
- The paper demonstrates strong empirical results on language modeling, exceeding the performance of other discrete diffusion models and approaching that of much larger continuous models. The image generation results, while not state-of-the-art, are compelling given the use of an off-the-shelf VQGAN tokenizer and suggest further potential with dataset-specific tokenizers.

**Weaknesses:**

- The model training only differs from absorbing discrete diffusion(MD4, MDLM) by using NCE-style loss vs cross-entropy based loss, and since the regular absorbing diffusion models assume factorized denoising model distribution, this seems to me that the training objective of proposed model do not differ from previous works by much. So can you use a regular absorbing diffusion model for the GGM style sampling?
- Similarly, the defined forward process can also be seen as a special case of D3PM, and it might have simpler/different forms if considering the continuous time limit, which is absent in current work.
- The use of Glauber dynamics, which updates one token at a time, necessitates long sampling sequences (T=4096). Exploring alternative sampling strategies or modifications to the dynamics that enable faster convergence could significantly improve efficiency. Analyzing the impact of varying sequence lengths *T* on both sample quality and computational cost would be valuable.
- Although GGM achieves respectable FID scores, a significant gap remains compared to state-of-the-art GANs and continuous diffusion models on image generation. The authors acknowledge the limitations of the generic VQGAN tokenizer and suggest exploring dataset-specific tokenizers. However, this reliance on VQGANs, even with improvements, might inherently restrict the model's ability to capture fine-grained details and diversity compared to methods operating directly in pixel space.

**Questions:**

Will the authors open-source the code for training and evaluating the proposed models?

---

> ### Author Response · Authors · 2024-11-17
> **Response to Reviewer cyno**
>
> We thank the reviewer for recognizing our methodological novelty and the strong empirical results obtained. We address the concerns raised below.
>
> > The model training only differs from absorbing discrete diffusion(MD4, MDLM) by using NCE-style loss vs cross-entropy based loss, and since the regular absorbing diffusion models assume factorized denoising model distribution, this seems to me that the training objective of proposed model do not differ from previous works by much. So can you use a regular absorbing diffusion model for the GGM style sampling?
>
> Our method fundamentally differs from absorbing diffusion in that we do not use an absorbing state (such as an explicit “mask” token). Our noising process replaces some position with a valid token from the vocabulary at all times. This is what fundamentally allows us to reduce the learning process to a binary classification. In case of an absorbing diffusion with an explicit mask token, such reduction is not possible. This also allows us to flip each token multiple times, enabling us to get multiple “versions” of the output before giving out the final version.
>
> As the reviewer points out, approaches like MDLM assume a factorized denoising model, whereas we do not make any of those assumptions. Moreover, MDLM models learn to predict $X_0$ by optimizing the variational lower bound. At a first glance, these two approaches do not seem compatible in terms of learning and sampling, without a significant redesign.
>
> > Similarly, the defined forward process can also be seen as a special case of D3PM, and it might have simpler/different forms if considering the continuous time limit, which is absent in current work.
>
> We have addressed the comparison to D3PM style forward processes in Remark 1. To summarize, considering this special case is what allows us to reduce the task of learning the denoising Markov chain to binary classification and helps us avoid making any assumptions (e.g., fully factored) on the distribution.
>
> We are not aware of a way to take the continuous limit of our presented forward process – as we have explicitly designed our backward process to be a discrete time Markov chain. This design choice was deliberate since prior works like D3PM make assumptions that the reverse process can be factorized in order to denoise multiple tokens simultaneously. This follows from the assumption of conditional independence of the reverse process, which is a stringent assumption. However, we do not require any such assumptions.
>
> > The use of Glauber dynamics, which updates one token at a time, necessitates long sampling sequences (T=4096). Exploring alternative sampling strategies or modifications to the dynamics that enable faster convergence could significantly improve efficiency. Analyzing the impact of varying sequence lengths T on both sample quality and computational cost would be valuable.
>
> We agree with the reviewer that exploring alternative sampling strategies or modifications to the dynamics to enable faster convergence could significantly improve efficiency. For example, something based on [1] can possibly be done. We leave this to future work since we expect this to require significant changes in the architecture and the algorithms.
> However, compared to prior work, we would like to point out that despite enabling parallel prediction, both SEDD and Plaid too require similar number of sampling steps to obtain good quality (e.g., T=4096 for Plaid and =2048 for SEDD).
>
> [1] Parallelising Glauber dynamics, Lee et. al., 2024
>
> > Although GGM achieves respectable FID scores, a significant gap remains compared to state-of-the-art GANs and continuous diffusion models on image generation. The authors acknowledge the limitations of the generic VQGAN tokenizer and suggest exploring dataset-specific tokenizers. However, this reliance on VQGANs, even with improvements, might inherently restrict the model's ability to capture fine-grained details and diversity compared to methods operating directly in pixel space.
>
> Operating directly in the pixel space is very inefficient, especially for generating high resolution images. Models operating in the latent space have been the industry standard [1, 2, 3] for the past few years, achieving state of the art quality. Therefore, we believe VQGAN based approaches can be scaled even in the case of discrete diffusion to obtain high quality images.
>
> [1] Imagen 3: https://storage.googleapis.com/deepmind-media/imagen/imagen_3_report.pdf
> [2] DALL-E 3: https://cdn.openai.com/papers/dall-e-3.pdf
> [3] Stable Diffusion 3.5: https://arxiv.org/pdf/2403.03206
>
> > Will the authors open-source the code for training and evaluating the proposed models?
>
> We are currently in the process of open-sourcing our code by obtaining necessary permissions.

---

### Official Review · Reviewer_zzst · 2024-11-04

**Soundness:** 3
**Presentation:** 3
**Contribution:** 2
**Rating:** 6
**Confidence:** 3

**Summary:**

The paper introduces the Glauber Generative Model (GGM), a new approach to generating discrete data like text and images using a process inspired by physics called Glauber dynamics. GGM works by changing one element at a time and simplifies the learning process to basic yes/no decisions about whether each element is meaningful or random noise. When tested on language generation, GGM performs better than similar existing models while using fewer computational resources, though it does not yet match the quality of top language models. For image generation, GGM produces good results without needing specialized preprocessing. While there is still room for improvement, the authors argue that GGM's straightforward approach makes it a promising direction for future research in generation systems.

**Strengths:**

This work introduces a theoretically elegant solution by reducing a complex generative task to simple binary classification problems. This makes the approach more straightforward to implement and understand compared to previous methods.

The model achieves strong empirical results, outperforming existing discrete diffusion models in language generation while using fewer parameters. It also shows competitive performance in image generation without requiring specialized tokenizers.

The method is versatile, working effectively across different domains (text and images).

The approach is computationally efficient, with lower complexity than previous methods, making it more practical to implement and scale.

**Weaknesses:**

Despite its improvements over other discrete diffusion models, GGM still performs worse than state-of-the-art methods like GPT models for text generation and continuous diffusion models for image generation.

The experimentation focuses mainly on comparing with other discrete diffusion models rather than providing comprehensive comparisons against the broader range of current generation methods.

The paper does not provide detailed analysis of computational costs and training times compared to other methods.

**Questions:**

How would the model perform if given more data and computing power?

How does the model perform on NLP tasks?

How stable is the training process?

How well does the model handle long-range dependencies in text?

Why don't you use the more common name "Gibbs sampler" or "time inhomogeneous Gibbs sampler"?

Is the classification-based learning method related to noise contrastive estimation?

---

> ### Author Response · Authors · 2024-11-17
> **Response to Reviewer zzst**
>
> We thank the reviewer for the recognizing the theoretical elegance, strong empirical results, and the versatility of our proposed method. We address the concerns raised below.
>
> > Despite its improvements over other discrete diffusion models, GGM still performs worse than state-of-the-art methods like GPT models for text generation and continuous diffusion models for image generation.
>
> The large literature on discrete diffusion models focuses on algorithmic improvements for discrete diffusion models to match the toplines (i.e., autoregressive models for language and continuous diffusion for images). We have discussed these aspects and the relevant literature extensively in the manuscript. This has enabled the continued improvement of discrete diffusion based methods over the last few years, helping it perform closer to the toplines. This is necessary since the topline approaches have benefitted from years of research and development whereas discrete diffusion based approaches are relatively new. We believe that our work is a significant step in this direction. We respectfully disagree that being behind the toplines is a significant weakness of our work.
>
> > The experimentation focuses mainly on comparing with other discrete diffusion models rather than providing comprehensive comparisons against the broader range of current generation methods.
>
> We focus on discrete diffusion in this work and evaluate against recent discrete diffusion models and autoregressive models for text, and against several kinds of models (autoregressive, discrete diffusion, continuous diffusion, flow-based models, autoencoder-based models, GANs) for images. It would be helpful if the reviewer can specify which current generation methods are missing.
>
> > The paper does not provide detailed analysis of computational costs and training times compared to other methods.
>
> We provide the computational cost and training times for our method in the supplementary material (Appendix H).
>
> > How would the model perform if given more data and computing power?
>
> We observe improved generative perplexities when scaling the model to more data (RedPajama) and parameters (~800M), leading to a generative perplexity of $16.7$ when evaluated using GPT2-xl.
>
> > How does the model perform on NLP tasks?
>
> Our focus in this paper is to come up with a foundation model relying on discrete diffusion. We demonstrate this by evaluating our model on unconditional text and image generation benchmarks. Achieving good performance on downstream NLP tasks usually requires a supervised finetuning stage. We leave this (and other interesting directions like instruction tuning of GGMs) to future work.
>
> > How stable is the training process?
>
> We observe that the training process remains stable throughout, with no loss spikes, and converges nicely. Like other discrete diffusion models (e.g., SEDD), we use EMA during inference.
>
> > How well does the model handle long-range dependencies in text?
>
> On qualitatively evaluating sequences of length 1024 generated by our model, we observe that the generated text remains on topic and doesn’t drift. We leave the extensions and evaluations of our model on larger sequence lengths to future work.
>
> > Why don't you use the more common name "Gibbs sampler" or "time inhomogeneous Gibbs sampler"?
>
> Due to their training, the authors are more familiar with probability theory literature. Here this Markov chain is popular as “Glauber dynamics” rather than “Gibbs Sampler”.
>
> > Is the classification-based learning method related to noise contrastive estimation?
>
> There are some high-level similarities between the two. For example, [1] use applies a NCE-style framework for language modeling. [1] optimizes binary cross-entropy to discriminate whether or not a token $w_{i+1}$ (given context $c_i=(w_1,\ldots,w_{i})$) arises from the model distribution or the noise distribution and is similar in structure to ours. However, our approach differs significantly from approaches like [1]. In terms of motivation, we use the reduction to binary classification to learn the denoising Markov chain while [1] uses the reduction to binary classification to learn $P(w_{i+1}|c_i)$, specifically to tackle large vocabulary sizes. [1] does not have any concept of time or gradual noising / denoising. Furthermore, the context used by [1] is still autoregressive (more specifically, $c_{i}$ only contains the tokens preceding $w_{i+1}$). [1] also requires very careful hyperparameter tuning with unconventional learning rate schedules to work well, and there is no evidence of it scaling to large datasets or parameters.
>
> [1] Improving Language Modelling with Noise Contrastive Estimation (Liza et. al., 2018)

---

> > ### Comment · Reviewer_zzst · 2024-11-25
> >
> > Thank authors for the detailed reply, which has addressed my points. I keep my positive rating.

---

### Official Review · Reviewer_tQ8L · 2024-11-05

**Soundness:** 3
**Presentation:** 3
**Contribution:** 3
**Rating:** 6
**Confidence:** 3

**Summary:**

This paper introduces a new approach to discrete diffusion models called the Glauber Generative Model. The key innovation is using the Glauber dynamics from statistical physics as a discrete Markov chain to denoise sequences of tokens. The authors reduce the denoising process to a series of binary classification tasks, making it simpler than previous approaches. They evaluate GGM on language and image generation tasks, showing it outperforms existing discrete diffusion models for language generation. It also demonstrates competitive performance on image generation without using dataset-specific tokenizers.

**Strengths:**

1. Good Physical Intuition: The use of Glauber dynamics in the paper is well-motivated and mathematically principled. Glauber dynamics serves as a discrete analog to Langevin dynamics, which underlies many continuous diffusion models, making it a natural and theoretically elegant choice for discrete diffusion.

2. Theoretical Innovation: The paper presents a novel theoretical framework that elegantly reduces the complex problem of discrete diffusion to simple binary classification tasks. This provides a more efficient approach compared to previous methods that required learning full transition matrices.

3. Plenty of Empirical Results: GGM outperforms existing discrete diffusion models on language generation tasks while using fewer parameters. For example, it achieves better generative perplexity than previous methods with fewer parameters and comparable performance. Besides, it also achieves strong performance for image generation without using dataset-specific image tokenizers.

**Weaknesses:**

1. Performance Gap: Despite improvements over other discrete diffusion models, the performance of GGM is still worse than state-of-the-art autoregressive models for language generation and continuous diffusion approaches.

2. Computational Requirements: The model requires a relatively large number of denoising steps compared to other approaches. While each step may be simpler, the total computational cost during generation could be large.

3. Limited Ablation Studies: The paper doesn't explore how different architectural choices and hyperparameters affect performance. For example, there's limited analysis of how the number of denoising steps or the choice of noise distribution impacts results.

**Questions:**

1. Could the author provide theoretical analysis or bounds on the convergence rate of GGM compared to standard Glauber dynamics?
2. Could the author give ablation studies about the number of denoising steps, different round-robin scheduling strategies for token positions and vocabulary size?
3. How do the dataset-specific tokenizers and general-purpose tokenizers affect the result of image generation?

---

> ### Author Response · Authors · 2024-11-17
> **Response to Reviewer tQ8L**
>
> We thank the reviewer for the recognizing the various strengths of our proposed method (good intuition, theoretical innovation, plenty of empirical evidence, etc.). We address the concerns raised below.
>
> > Performance Gap: Despite improvements over other discrete diffusion models, the performance of GGM is still worse than state-of-the-art autoregressive models for language generation and continuous diffusion approaches.
>
> The large literature on discrete diffusion models focuses on algorithmic improvements for discrete diffusion models to match the toplines (i.e., autoregressive models for language and continuous diffusion for images).  We have discussed these aspects and the relevant literature extensively in the manuscript. Such an approach has enabled the continued improvement of discrete diffusion based methods over the last few years, helping it perform closer to the toplines. We believe that our work is a significant step in this direction. We respectfully disagree that being behind the toplines is a significant weakness of our work.
>
> > Computational Requirements: The model requires a relatively large number of denoising steps compared to other approaches. While each step may be simpler, the total computational cost during generation could be large.
>
> We compare the computational complexity of our method with other discrete diffusion based approaches and with autoregressive models separately below.
>
> **Comparison to other discrete diffusion approaches:** Approaches that predict multiple tokens in parallel need not predict all those tokens correctly. Parallel prediction can lead to errors being produced at multiple positions at once. We believe that this could be the reason why parallel denoising methods like Plaid and SEDD still require the same order of sampling steps ($T=4096$ for Plaid and $T=2048$ for SEDD) as us and still do not achieve better generation quality. Note that compared to these methods – which allow each token to be flipped $T$ times ($T=4096$ for Plaid and $T=2048$ for SEDD) – our method still does better while giving each token only $4$ chances to be changed.
>
> In future work, we hope to explore alternative architectures and other recently proposed approaches (such as [1]) for parallelizing Glauber dynamics to improve the efficiency while maintaining quality.
>
> [1] “Parallelising Glauber dynamics” (Lee et al. 2024)
>
> **Comparison to Autoregressive models:** The main algorithmic advantage of our model is that it can parse the generated output multiple times and fix it with multiple noisy versions. This can allow complex generations and planning – which is one of the fundamental motivations for considering discrete diffusion based approaches [1, 2]. Discrete diffusion models are known to outperform autoregressive models in many settings such as 3-SAT and Sudoku as demonstrated in the literature. Thus, we believe that a straightforward comparison of runtime with autoregressive models is not fruitful with this motivation in mind.
> However, we agree that several works on discrete diffusion models are aimed at flipping multiple tokens at the same time, in order to reduce computational complexity compared to autoregressive models. As we note above, SoTA discrete diffusion approaches still end up requiring similar number of forward passes as our model to achieve good quality.
>
> [1] LayoutDM: Discrete Diffusion Model for Controllable Layout Generation (Inoue et. al., 2023)
> [2] Beyond Autoregression: Discrete Diffusion for Complex Reasoning and Planning (Ye et. al., 2024)
>
> > Limited Ablation Studies: The paper doesn't explore how different architectural choices and hyperparameters affect performance. For example, there's limited analysis of how the number of denoising steps or the choice of noise distribution impacts results. Could the author give ablation studies about the number of denoising steps, different round-robin scheduling strategies for token positions and vocabulary size?
>
> Experiments to try out different denoising steps and noising distributions require pre-training from scratch, which is very expensive at the 400M parameter scale. We are currently bottlenecked by the resources necessary for this but we hope to explore these ablations in a future version.
>
> > How do the dataset-specific tokenizers and general-purpose tokenizers affect the result of image generation?
>
> Data specific tokenizers are known to improve the generation quality by a lot, as pointed out in prior works. We have discussed this in Section 4.2 under “Comparison with SoTA methods”, with relevant references. We believe that exploring methods to improve image tokenization is beyond the scope of our present work but is a great avenue for future research.

---

> ### Author Response · Authors · 2024-11-17
> **Response to Reviewer tQ8L (contd.)**
>
> > Could the author provide theoretical analysis or bounds on the convergence rate of GGM compared to standard Glauber dynamics?
>
> Standard Glauber dynamics cannot be compared effectively with GGM as is. Standard Glauber Dynamics is an MCMC algorithm which assumes the knowledge of the Hamiltonian (i.e., log probability up to additive constants) whereas GGM learns to produce more samples given some samples from the target distribution.
>
> Below, we compare them in the following setting:
>
> - Algorithms A: The neural network learns the transition probabilities of the standard Glauber dynamics from data. The learning is perfect (i.e, zero error)
> - Algorithm B: GGM (our algorithm). The learning is perfect (i.e, zero error)
>
> Algorithm A is known to take exponential time in $L$ to converge to the target distribution even in simple settings (see [1], [2] below).
>
> As for Algorithm B: Suppose we consider $\Pi_t = \Pi$ to be the distribution where $\Pi(\phi) = 0.5$ (as is considered in the experimental evaluations). Combining Lemmas 1, 2 and 3 in the paper with $T = O(L\log (L/\delta))$, we can show that $ \text{TV}(\text{Law}(X_0),P^*) \leq \delta$ for any $\delta \in (0,1)$. We will include this proof in the revised version of our paper.
>
> [1] Markov Chains and Mixing Times (Levin el. al., see chapter 15.6)
> [2] Mixing time of exponential random graphs (Bhamidi et. al., 2008)
>
> This is analogous to the case of continuous diffusion models where time-invariant Langevin dynamics can fail to sample mixture of Gaussians accurately in polynomial time whereas DDPM/ diffusion models can accurately sample these distributions ([3],[4])
>
> [3] Sampling Is as Easy as Learning the Score: Theory for Diffusion Models With Minimal Data Assumptions
> [4] Learning Mixtures of Gaussians Using the DDPM Objective

---

### Official Review · Reviewer_8Yq4 · 2024-11-09

**Soundness:** 3
**Presentation:** 3
**Contribution:** 3
**Rating:** 6
**Confidence:** 3

**Summary:**

The paper proposes a new type of discrete diffusion model. The denoising process is defined as flip one token at a time, and the model predicts whether the token at the given location is noise. The proposed GGM is tested on both text and image, and shows strong generative perplexities compared with discrete diffusion model baselines.

**Strengths:**

- The method is innovative and yields promising results.
- The paper is clearly written, with a thorough discussion of related work.
- The experiments are comprehensive and well-documented.

**Weaknesses:**

Clarification questions:
- In Algorithm 2, line 5, the step is executed "for all a." Does this require $|\mathcal{X}|$ forward passes per model? If so, could this impact efficiency? Is there a way to parallelize this?
- Why can the model take a mask token as input? According to Eq. (1), $X_{t+1}$ inherits $X_t$ when $Z_t = \phi$. When exactly does $Z_t$ take the value $\phi$?

**Questions:**

Please see Weakness clarification questions.

---

> ### Author Response · Authors · 2024-11-17
> **Response to Reviewer 8Yq4**
>
> We thank the reviewer for recognizing our innovative method, clear writing, and the comprehensive experiments yielding promising results. We have addressed the questions raised by the reviewer below and hope that they would consider re-evaluating their score.
>
> > In Algorithm 2, line 5, the step is executed "for all a." Does this require |\mathcal{X}| forward passes per model? If so, could this impact efficiency? Is there a way to parallelize this?
>
> We parameterize our model using a special mask token $\omega$ which allows us to compute all $|\mathcal{X}|$ logits in parallel, in a single forward pass. Conditioning on the token ‘$a$’ comes by virtue of selecting only the logit corresponding to that token, instead of re-evaluating the entire model for every token with the token at the $i^{th}$ position.
>
> > Why can the model take a mask token as input? According to Eq. (1), $X_{t+1}$ inherits $X_t$ when $Z_t=\phi$. When exactly does $Z_t$ take the value $\phi$?
>
> As explained earlier, we parameterize our architecture with a special mask token $\omega$ which allows us to compute all $|\mathcal{X}|$ logits in parallel, in a single forward pass. This token is unrelated to $Z_t$ and $\phi$ and only serves as an architectural design choice.
>
> We refer to Section 3.1, where the random variable $Z_t$ is described. $Z_t$ is a random variable over the set $\\{\phi\\}\cup \mathcal{X} $ and $\Pi_t$ is a distribution over this extended set $\\{\phi\\}\cup \mathcal{X}$. An example of a distribution $\Pi_t$ is $\Pi_t(Z_t = \phi) = 0.5$ and $\Pi_t(Z_t=a)=0.5/|\mathcal{X}|$ for all $a \in \mathcal{X}$.

---

> > ### Comment · Reviewer_8Yq4 · 2024-11-25
> > **thanks for the rebuttal**
> >
> > I thank the authors for the rebuttal. I will keep my rating positive. I am curious how is this parallelism implemented? Is it like the projection discriminator, i.e. calculate the inner product of mask token output embedding and class token embedding?

---

> > > ### Author Response · Authors · 2024-11-26
> > >
> > > **Method to Parallelize**:
> > >
> > > Our parallelization process is much more basic than the one proposed by the reviewer (we hope to explore more sophisticated architecture in future work). During the learning process, the logit corresponding to the element 'a' learns to output the probability $P(Z_t = 1| X_{t+1,-i} = x_{-i}, X_{t+1,i} = a)$ but with the input $x_{-i} = X_{t+1,-i},$ with masked position $i$ (i.e, $x_i = \omega$). During inference, each logit 'a' inherently assumes that position $i$ is 'a' and outputs the respective posterior probability for $Z_t$. Since we evaluate all the logits with a single forward pass like in a typical transformer architecture, we achieve paralleism.
> > >
> > > **More information about the learning procedure:**
> > >
> > > The output of logit 'a' is trained only over examples where $X_{t+1,i} = a$ but with the masked input $x_{-i}, x_i = \omega$. That is, when we obtain a sample with $X_{t+1,-i} = a$, we input the masked token in position $i$ and we do not train the logit outputs corresponding to other elements $b \neq a$. Due to this, it learns to predict the output assuming inherently that the input had $X_{t+1,i} = a$ (due to the distribution of input). We can show via the properties of the binary cross entropy loss that the minimizer of our training loss function is when logit 'a' outputs  $P(Z_t = 1| X_{t+1,-i} = x_{-i}, X_{t+1,i} = a)$.

---

> > > > ### Comment · Reviewer_8Yq4 · 2024-11-26
> > > > **thanks for the clarification**
> > > >
> > > > I appreciate the authors’ clarification; it is now much clearer. I think they are equivalent, as the last linear layer effectively computes inner products for all classes $emb(\omega)\cdot emb_a$ in parallel. And I think including this explanation in the revision would enhance both readability and clarity.

---

> > > > > ### Author Response · Authors · 2024-11-26
> > > > >
> > > > > Thank you for the connection and the suggestion. We are happy to include this explanation in the revised version.

---

### Meta-Review · Area_Chair_vAi5 · 2024-12-22

**Metareview:**

This paper proposes a novel discrete diffusion method by casting the denoising step as binary classifications problems. It shows strong results on both language and quantized image modeling tasks. Reviewers overall find the method interesting and elegant, and the results convincing. The most common critique is that this method still lags behind SOTA results, ie that from a strong autoregressive model, but the AC believes that this is a understandable gap and shouldn't take way the main contributions of the paper, improving discrete diffusion alone is sufficient to justify a standalone contribution. Other concerns, like its relationship to absorbing state discrete diffusion models are addressed by authors in the rebuttal, and the AC believes that they are not reasons to reject this work. Overall, the AC believes that this is a solid work and recommend accept.

**Additional Comments On Reviewer Discussion:**

The authors addressed most of the clarification questions in the rebuttal.

---

### Decision · Program_Chairs · 2025-01-22

Accept (Poster)